# Characterization of a novel subfamily 1.4 lipase from *Bacillus licheniformis* IBRL-CHS2: Cloning and expression optimization

**Ammar Khazaal Kadhim Almansoori**[1,2], **Nidyaletchmy Subba Reddy**[1], **Mustafa Abdulfattah**[1,3], **Sarah Solehah Ismail**[1], **Rashidah Abdul Rahim**[1,3]*

**1** School of Biological Sciences, Universiti Sains Malaysia, Gelugor, Penang, Malaysia, **2** Department of Medical Laboratory Techniques, Al-Mustaqbal University, Hillah, Babylon, Iraq, **3** Centre for Chemical Biology (CCB), Universiti Sains Malaysia, SAINS@USM, Bayan Lepas, Penang, Malaysia

* rshidah@usm.my

## Abstract

This study focuses on a novel lipase from *Bacillus licheniformis* IBRL-CHS2. The lipase gene was cloned into the pGEM-T Easy vector, and its sequences were registered in GenBank (KU984433 and AOT80658). It was identified as a member of the bacterial lipase subfamily 1.4. The pCold I vector and *E. coli* BL21 (DE3) host were utilized for expression, with the best results obtained by removing the enzyme's signal peptide. Optimal conditions were found to be 15°C for 24 h, using 0.2 mM Isopropyl β-D-1-thiogalactopyranoside (IPTG). The His-tagged lipase was purified 13-fold with a 68% recovery and a specific activity of 331.3 U/mg using affinity purification. The lipase demonstrated optimal activity at 35°C and pH 7. It remained stable after 24 h in 25% (v/v) organic solvents such as isooctane, n-hexane, dimethyl sulfoxide (DMSO), and methanol, which enhanced its activity. Chloroform and diethyl ether inhibited the lipase. The enzyme exhibited the highest affinity for *p*-nitrophenol laurate (C12:0) with a $K_m$ of 0.36 mM and a $V_{max}$ of 357 µmol min$^{-1}$ mg$^{-1}$. Among natural oils, it performed best with coconut oil and worst with olive oil. The lipase was stable in the presence of 1 mM and 5 mM $Ca^{2+}$, $K^+$, $Na^+$, $Mg^{2+}$, and $Ba^{2+}$, but its activity decreased with $Zn^{2+}$ and $Al^{3+}$. Non-ionic surfactants like Triton X-100, Nonidet P40, Tween 20, and Tween 40 boosted activity, while Sodium Dodecyl Sulfate (SDS) inhibited it. This lipase's unique properties, particularly its stability in organic solvents, make it suitable for applications in organic synthesis and various industries.

## Introduction

Chemical catalysis, while offering higher yields, faces challenges in product recovery and catalyst reusability. Heterogeneous and enzymatic catalysis present viable alternatives, reducing overall process costs [1]. Enzymatic catalysis, honed over billions of years in biological systems, offers green, mild, and efficient reactions. Biocatalysts, with their environmental friendliness

**Data Availability Statement:** All relevant data are within the paper and its Supporting Information files.

**Funding:** This work was supported by the Malaysian Ministry of Higher Education under the Fundamental Research Grant Scheme (FRGS/1/2023/STG02/USM/02/7). The grant has been awarded to Rashidah Abdul Rahim.

**Competing interests:** The authors declare that we do not have any conflict of interest to the content of this article.

and high activity, prevent side reactions and exhibit stereoselectivity in organic transformations, offering promising solutions [2].

Lipases, crucial enzymes in both biological processes and industrial applications, play versatile roles in catalyzing hydrolysis and synthesis reactions, particularly cleaving ester bonds between fatty acids and glycerol backbones [3]. They catalyze the breakdown of triacylglycerols (TAGs) into diacylglycerols (DAGs), monoacylglycerols (MAGs), glycerols, and fatty acids [4]. Widely distributed among animals, plants, and microorganisms, microbial lipases are particularly advantageous due to ease of cultivation and genetic manipulation [5]. Notably, lipases exhibit substrate selectivity, stability, and temperature adaptability, making them valuable biocatalysts for various industries including pharmaceuticals, detergents, food processing, and biodiesel. Their role is extensive and reusable, thus making it more efficient in a variety of industrial applications [6].

Messaoudi et al. presents a comprehensive database dedicated to 'true' lipases, which are a specific group of enzymes characterized by their ability to hydrolyze ester bonds in lipids. The study distinguishes between two main groups of lipases: carboxylesterases (EC 3.1.1.1) and 'true' lipases (EC 3.1.1.3), focusing on the unique features of 'true' lipases, such as interfacial activation, which is essential for their enzymatic function. This activation is linked to a consensus sequence and a conserved domain within the enzymes. LIPABASE, a centralized resource, compiles extensive information on 'true' lipases, including enzyme properties, sequences, and structural data, to help researchers explore the relationship between structure and function and support studies in enzymology and bioinformatics [7].

*Bacillus* and *Geobacillus* lipases hold significant interest owing to their biotechnology potential and extensive application across diverse industries. *Bacillus* lipases, including those of *B. subtilis* and other species, have been extensively studied and isolated for their enzymatic potential [8]. On the other hand, *Geobacillus* lipases, such as *G. zaliha* and *G. thermocatenulatus* have been characterized for their unique properties including thermal stability and pH-thermal optimization maintaining activity at 35°C [9]. Studies have shown that *Geobacillus* lipases can exhibit ion-pair interactions and hydrogen bonds that contribute to their thermal stability, making them valuable in industrial applications [10].

The conserved pentapeptide sequences of the I.4 and I.5 families of lipases, including *Bacillus* and *Geobacillus* lipases are essential for their structure and function these conserved pentapeptide sequences contain alanine residues that replace a precursor glycine [11]. This structural feature is shared by various *Geobacillus* lipases, suggesting a common evolutionary origin and functional similarity in this group of enzymes [12]. The structural rigidity and protein thermostability of various lipase A from *Bacillus subtilis* have been investigated, demonstrating the relationship between structural rigidity and thermostability [13].

*Bacillus subtilis* lipases with a molecular mass of 19.5 kDa are among the smallest known lipases and lack a lid domain [14]. LipA, a representative of these lipases, is termed "lidless" due to the absence of a conventional lid domain covering its active site [15]. Unlike other structurally similar lipases, polyethylene terephthalate (PET) hydrolases that degrade PET generally do not have a protective envelope and have a more pronounced and simpler active site [16]. Moreover, small lipases are known for their stability and ease of genetic manipulation and are useful for biotechnology applications [17].

The main objective of this research is to explore and investigate the expression and characteristics of the second subfamily 1.4 lipase derived from *Bacillus licheniformis*. While the first subfamily 1.4 lipase from this bacterium, referred to as LipA in this study, has been extensively studied [18,19], Rey et al. (2004) identified a second subfamily 1.4 lipase in *Bacillus licheniformis* during a genome sequencing project [20]. Despite this discovery, the properties of this second lipase remained unexplored until Jo et al. (2014) used *E. coli* to produce a whole-cell

biocatalyst expressing this enzyme [21]. Although they successfully expressed the enzyme on the surface of *E. coli*, a detailed characterization of its properties was not performed. As such, further investigation is needed to determine whether this lipase exhibits unique properties or differs from other members of the subfamily.

Furthermore, Reddy et al. (2016) attempted to clone the *Bacillus licheniformis* lipase using the pET-15b (+) vector but did not succeed in expressing or characterizing the enzyme [22]. This study addresses this limitation by utilizing the pCold I vector system and removing the signal peptide, which enabled successful expression and detailed characterization. This approach revealed distinct properties of the enzyme, such as stability in organic solvents and enhanced activity in non-ionic surfactants, features not previously reported for this class of enzymes.

The discovery of enzymes with novel and unique properties is critical to meet the increasing demands of industrial applications. This study introduces a previously reported lipase from *Bacillus licheniformis* IBRL-CHS2, a member of bacterial lipase subfamily 1.4, and demonstrates its stability in organic solvents, enhanced activity in non-ionic surfactants, and optimal expression at low temperatures. These unique characteristics distinguish this enzyme from other known lipases, including those studied by Reddy et al. (2016) [22]. The originality of this research lies in the enzyme's distinct properties and its potential for industrial applications, particularly in challenging environments where enzyme stability and activity are paramount.

## Materials and methods

### Microorganism

The *Bacillus licheniformis* IBRL-CHS2 strain utilized in this study was acquired from the Industrial Biotechnology Research Laboratory (IBRL) at Universiti Sains Malaysia (USM) and was provided as a glycerol stock. The bacterium was previously isolated and identified as *Bacillus licheniformis* through 16S rDNA sequencing, as detailed by Noor Mazuin et al. [23].

### Construction of pCold-MLipA

To extract genomic DNA from the *Bacillus licheniformis* IBRL-CHS2 culture, the Cetyltrimethylammonium Bromide (CTAB) extraction method described by Doyle (1991) was employed [24]. Following DNA extraction, amplification of the lipase gene was conducted using specific forward (BLF) and reverse (BLR) primers (designated as S1 Table). The lipase gene sequence from *Bacillus licheniformis* ATCC 14580 (DSM 13), with GenBank Accession number AE017333, was obtained from the National Center for Biotechnology Information (NCBI) database. For the construction of MLipA, the lipase gene was cloned into the pCold-I vector (Takara, Clontech, Japan) using the NdeI and BamHI restriction sites.

### Expression and purification of MLipA

Protein expression was initiated by inducing a prechilled bacteria, *E. coli* BL21 (DE3), pCold-MLipA, with 0.2 mM IPTG for 30 min after reaching Obtical Density $OD_{600}$ ~ 0.5. The induced cultures were then incubated at 15˚C and shaken at a speed of 150 rpm for a duration of 16 h. His-tagged immobilized metal affinity chromatography (IMAC) was utilized to purify the lipase using the HisTALON™ Gravity Column Purification Kit (Takara, Clontech, Japan), following the manufacturer's instructions. Subsequently, the purified lipase was subjected to sodium dodecyl sulfate-polyacrylamide gel electrophoresis (SDS-PAGE) to verify its purity and molecular weight. The concentration of the purified lipase was assessed utilizing the Bio-

Rad Protein Assay, a widely employed method that follows the Bradford assay principle (Bio-Rad, USA) [25].

**Lipase assay.** Lipase activity was evaluated using a colorimetric technique, adapting the method described by Ch'ng & Sudesh with minor adjustments [26]. To prepare the substrate, 5 mM of p-nitrophenyl laurate (p-NPL) (Sigma, USA) was dissolved in 10 mL of DMSO (Sigma, USA) immediately before each assay. The substrate solution was then emulsified by combining 1 mL of the solution with 9 mL of 0.1 M phosphate buffer (pH 7), which contained 0.1% (w/v) polyvinyl alcohol (Fluka, USA) and 0.01% Triton X-100. For the assay, 180 μL of the substrate mixture was added to each well of a 96-well plate and allowed to equilibrate at 35°C for 5 minutes. Afterward, 10 μL of lipase solution (5 μg/mL) was added. A blank control was also prepared under the same conditions, except 10 μL of 0.1 M phosphate buffer (pH 7) was used in place of the lipase. The reaction was stopped after a 10-minute incubation at 35°C by adding 10 μL of 6 M HCl, and the absorbance was recorded at 410 nm using a Multiskan GO Microplate Spectrophotometer (Thermo Scientific, USA). All experiments were performed in triplicate, and the data were expressed as the mean ± standard deviation (SD). Lipase activity was calculated based on the amount of p-nitrophenol produced. One unit of lipase activity is defined as the quantity of enzyme required to release 1 μmol of p-nitrophenol per minute. The equations used for calculating lipase activity and specific activity are provided below:

$$\text{Lipase activity (U/mL)} = \frac{\text{amount of } p-\text{nitrophenol released } (\mu\text{mol})/\text{min}}{\text{volume (mL)}}$$

$$\text{Lipase specific activity (U/mg)} = \frac{\text{lipase activity (U/mL)}}{\text{lipase concentration (mg/mL)}}$$

## Characterization of recombinant MLipA

**Influence of temperature on the activity and stability of MLipA** The MLipA activity was evaluated across a range of temperatures (15°C– 50°C) in order to determine its optimum temperature. The lipase assay involved the equilibration of the substrate at the desired temperature, followed by the initiation of the reaction. To assess the temperature stability of MLipA, the lipase was incubated in a 0.1 M phosphate buffer at pH 7 for a duration of 1 h at various temperatures. Subsequently, the residual activity of the lipase was measured at its optimum temperature. Additionally, the lipase was incubated for 24 h at temperatures of 30°C, 35°C, and 40°C. During the incubation period, samples were collected at different time intervals and subsequently subjected to activity assays at the lipase's optimum temperature. The relative activity and residual activity of the lipase at each temperature were calculated using the equations or formula given in S1 File.

## Influence of pH on the activity and stability of MLipA

The purified LipA activity was tested across a range of pH levels from 4 to 11, all while maintaining a constant temperature of 35°C to find its best pH. Various types of buffers were utilized for the lipase test, including acetate buffer at pH 4 and pH 5, sodium phosphate buffer (pH 6 and pH 7), glycine-NaOH buffer (pH 9 and pH 10), pH 8 of Tris-HCl buffer and carbonate buffer (pH 11). 10 μL of 1M sodium carbonate was added to terminate the lipase activity. Stability of the lipase under different pH conditions was also examined by leaving it in the corresponding buffers for 1 h and then measuring its remaining activity (%). Moreover, stability was assessed in different pH including pH 6, pH 7, and pH 8 by leaving it in those buffers

for 24 h. During this time, samples were periodically taken, their activity was tested at 35°C, and the remaining activity (%) was calculated.

**Influence of organic solvents on the activity of MLipA.**   To evaluate their effect on lipase activity, three volumes of pure lipase (5 μg/mL) were combined with an equal amount of different 25% (v/v) organic solvents and incubated for one h and twenty-four h at 35°C and 150 rpm [27]. The log $P$ values of the organic solvents that were utilized varied. Diethyl ether (log $P$ = 0.85), chloroform (log $P$ = 2), n-hexane (log $P$ = 3.6), n-heptane (log $P$ = 4.27), and isooctane (log $P$ = 4.5) were among the hydrophobic organic solvents. Acetone (log $P$ = -0.23), acetonitrile (log $P$ = -0.15), ethanol (log $P$ = -0.24), methanol (log $P$ = -0.76), and DMSO (log $P$ = -1.22) were the hydrophilic organic solvents.

**Substrate specificity of MLipA.**   Using various $p$-nitroyl ($p$-NP) fatty acyl esters as substrates under standardized assay conditions, the substrate specificity of the purified lipase was investigated. These substrates included $p$-nitrophenyl octanoate (C8:0), $p$-nitrophenyl decanoate (C10:0), $p$-nitrophenyl laurate (C12:0), $p$-nitrophenyl myristate (C14:0), $p$-nitrophenyl palmitate (C16:0), and $p$-nitrophenyl stearate (C18:0) (Sigma, USA). The lipase's relative activity (%) toward each substrate was calculated by comparing it to the control, which was the lipase activity with $p$-nitrophenyl laurate.

To evaluate the ability of MLipA to hydrolyze natural oils, various oils such as coconut oil, palm oil, olive oil, and canola oil were utilized as substrates. The lipase assay followed the modification colorimetric method outlined by Kwon & Rhee [28].

**Determination of kinetic parameters of MLipA.**   The kinetic parameters of recombinant lipase MLBL was evaluated with the use of $p$-nitrophenyl laurate as a substrate. Various concentrations of the substrate, ranging from 0.1 mM to 1 mM, were prepared for lipase assay to measure the reaction velocity (V). The velocity of reaction was determined through the measurement of absorbance at a specific wavelength. The reciprocal values of the substrate concentration were plotted against the reciprocal values of the reaction velocity to generate a Lineweaver-Burk plot, which is a double reciprocal plot. The intercept on the y-axis represents the reciprocal of the maximum velocity ($V_{max}$), while the x-intercept represents the negative reciprocal of the Michaelis constant ($K_m$). The $V_{max}$ and $K_m$ values were calculated from the Lineweaver-Burk plot as indicators of the enzymatic efficiency and substrate affinity of the MLipA, respectively.

**Influence of metal ions on the activity of MLipA.**   To investigate the effect of different metal ions on lipase activity, the purified lipase (5 μg/mL) was incubated at 35°C at final concentrations of 1 mM and 5 mM of metal ions for 1 hour, following the protocol established by Rahman, et al. [29]. The control reaction, which did not contain any metal ions, was prepared by dialyzing the lipase against the buffer to remove any trace metal ions. The lipase activity in the presence of each metal ion was compared to the control reaction to calculate the relative activity (%).

**Influence of effector molecules on the activity of MLipA.**   The impact of different effector molecules on lipase activity was analysed through the 1 h incubation of purified lipase (5 μg/mL) with different effector molecules at final concentrations of 1 mM and 5 mM at 35°C, as described by Rahman, et al. [29]. Effector molecules included oxidizing agents like ammonium persulfate and potassium iodide, reducing agents such as ascorbic acid and 2-mercaptoethanol, metal chelating agents like EDTA and sodium citrate, and phenyl methyl sulphonyl fluoride (PMSF) as a serine hydrolase inhibitor. A control reaction lacking effector molecules was also conducted. Lipase activity in the presence of effector molecules was compared to the control reaction, and relative activity (%) was calculated accordingly.

**Influence of surfactants on the activity of MLipA.**   To evaluate how surfactants affect the activity of lipase, non-ionic surfactants (Tween-20, Tween-40, Tween-80, Triton X-100, Span-

40, and Nonidet P40) (Thermo Scientific, USA) at a concentration of 1 mM, along with the ionic surfactant SDS, were studied. Purified lipase (5 µg/mL) was mixed with surfactants in equal volumes and incubated for about 30 min at 35˚C before the lipase assay, following the method described by Prazeres and his colleagues [30]. A control reaction devoid of surfactant was included. Lipase activity in the presence of each surfactant was compared to the control, and relative activity (%) was calculated to evaluate their effect on lipase activity.

## Statistical analysis

All experiments in this study were conducted in triplicate to ensure reproducibility and minimize variability. The results are presented as mean ± standard deviation (SD), representing the dispersion of the data around the mean. The R-values (correlation coefficients) were calculated to assess the linearity and reliability of experimental data, ensuring consistency across measurements.

## Results

### Cloning of lipase gene from *Bacillus licheniformis*

Genomic DNA from *B. licheniformis* IBRL-CHS2 was extracted utilising a modified CTAB method, as described by Doyle in 1991. The lipase gene was amplified via Polymerase chain reaction (PCR) using specific primers, resulting in a 615 bp product (Fig B in S2 Fig). Gel electrophoresis of the PCR products revealed two distinct bands, approximately 2,500 bp and 615 bp. The 615 bp lipase gene band was subsequently purified (Fig C in S2 Fig), while the larger 2,500 bp band was identified as a nitrite transporter gene after sequencing.

The LipA gene was cloned into a pGEM-T easy vector, and white colonies were analyzed through colony PCR, plasmid extraction, digestion, and DNA sequencing to confirm the presence of the lipase gene. Five white colonies were selected for colony PCR, resulting in successful amplification of the insert (Fig D in S2 Fig). One colony was cultured in LB broth supplemented with ampicillin, and the extracted plasmid was digested with EcoRI to confirm the presence of LipA (Fig E in S2 Fig). Subsequently, the purified lipase gene and pCold™ I vector were digested with NdeI and BamHI enzymes, ligated to clone the gene into the vector, and transformed into *Escherichia coli* cells. Colony PCR and sequencing confirmed the presence of the lipase gene and the vector.

### Sequence homology and phylogenetic analysis

The plasmid pGEM- LipA was subjected to sequencing at First Base Laboratories Sdn. Bhd. Analysis of the gene sequence (Fig 1A) using Blast indicated a high degree of similarity to the lipase gene from *B. licheniformis* DSM 13 = ATCC 14580, showing 99% homology. This lipase gene is classified within the alpha/beta hydrolases superfamily. The LipA peptide is composed of 204 amino acids with a predicted isoelectric point (pI) value of 9.43 and a molecular weight of 21.8 kDa. The signal peptide cleavage site for recombinant LipA is anticipated to be located between Ala-30 and Ala-31 (refer to S1 Fig). The pI of MLipA, or the mature lipase, has a predicted value of 9.05 and 18.5 kDa molecular weight. The GenBank accession numbers for the nucleotide and amino acid sequences of LipA are KU984433 and AOT80658, respectively.

Comparative analysis of LipA with other *Bacillus* lipases revealed 65% homology to LipA from *B. licheniformis* (NCBI accession no. CAB95850) and 66% homology to LipA from *B. licheniformis* (ACB38749). Similarly, LipA demonstrated 69% homology to the lipase of *B. amyloliquefaciens* (AGO17775), and 67% homology to both LipA from *B. subtilis* (AAA22574) and the lipase from *B. pumilus* (AAR84668). Despite variations in signal peptide sequences,

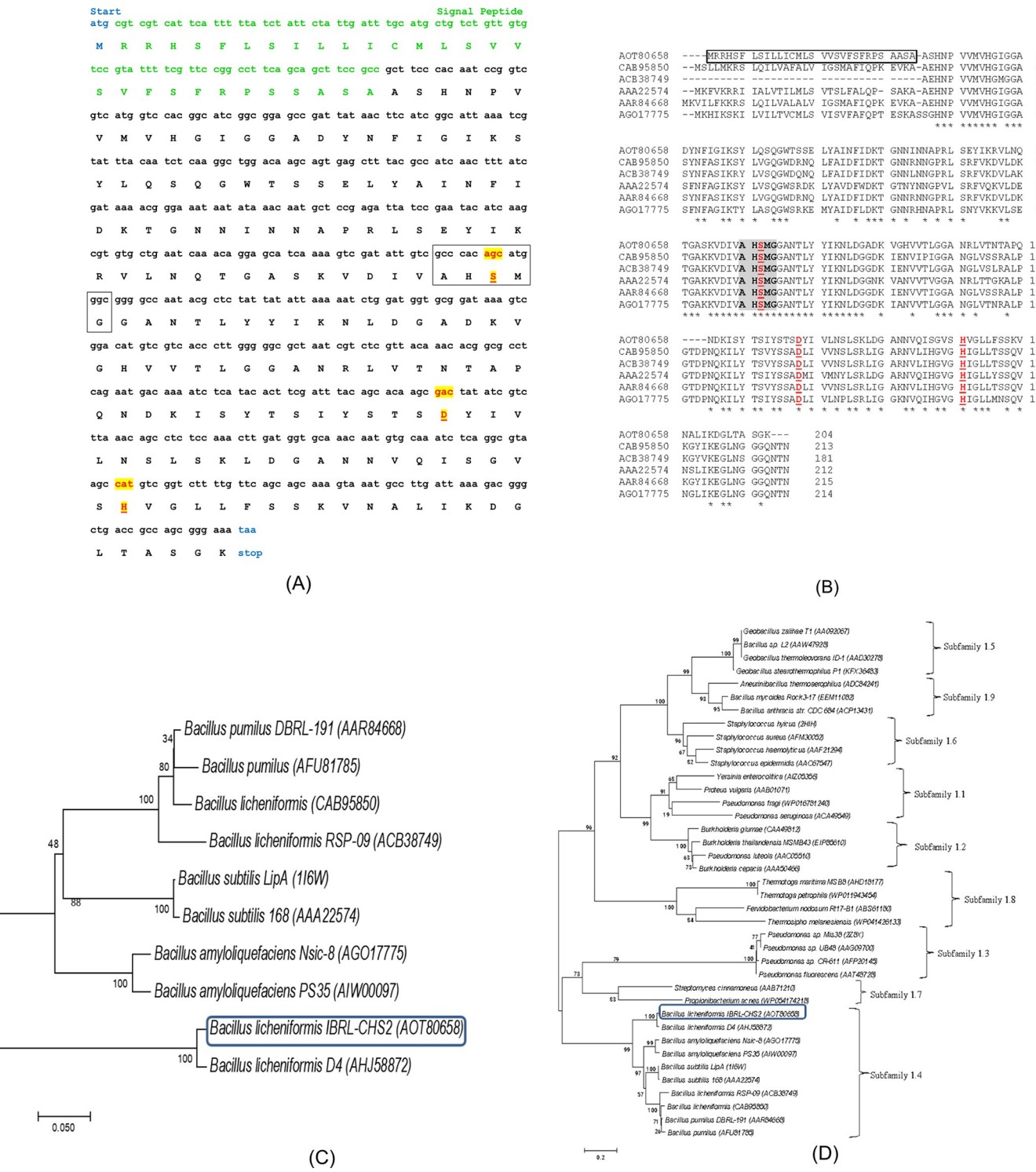

**Fig 1. Molecular characterization and phylogenetic analysis of LipA_B.licheniformis.** (A) shows the nucleotide and amino acid sequences of LipA_B.licheniformis. The signal peptide is identified in green, and the start and stop codons are indicated in blue. The conserved pentapeptide sequence of LipA_B.licheniformis is enclosed in a box, and the inferred catalytic residues Ser, Asp, and His are highlighted in yellow and labeled in red. (B) displays the amino acid sequence alignment of LipA_B.licheniformis (NCBI accession number: AOT80658) with lipases from other *Bacillus* sp. belonging to subfamily 1.4. These include *Bacillus licheniformis* (CAB95850), *Bacillus licheniformis* RSP-09 (ACB38749), *B. subtilis* Lip A (AAA22574), *B. pumilus* (AAR84668), and *B. amyloliquefaciens* (AGO17775). The conserved pentapeptide is highlighted in grey and shown in bold. The putative catalytic triad residues, Ser, Asp, and His, are denoted in red and underlined. Similar amino acids in the sequences are denoted with asterisks (*). The signal peptide for LipA_B.licheniformis is indicated within a box.

alignment of amino acid sequences of these lipases highlighted the preservation of the penta-peptide box and catalytic residues. The catalytic triad of LipA, identified through alignment with other *Bacillus* lipases (Fig 1B) and this investigation, includes Ser-107, Asp-159, and His-182 (corresponding to Ser-77, Asp-129, and His-152 in the mature lipase).

## Expression and purification of recombinant MLipA

Upon successful cloning of the lipase gene (525 bp) into the pCold^TM I vector, the expression of lipase was activated using the cold-shock protein A (cspA) promoter and T7 promoter. Subsequently, crude soluble lipase was obtained through sonication, as illustrated in SDS-poly-acrylamide gel (Fig 2). Achieving the successful expression of recombinant LipA required optimizing both the vector and host systems. Initially, several vector-host combinations were tested but failed to produce sufficient quantities of biologically active recombinant LipA. Consequently, the gene sequence of LipA was modified by removing the signal peptide coding sequence, while ensuring its functional domain remained unchanged and unaltered. This modification led to the expression of LipA using the pCold I vector in *E. coli* BL21 (DE3). This system expressed MLipA as inclusion bodies (Fig 2A and 2B).

The pCold-MLipA *E. coli* BL21 (DE3) system proved to be more stable and reliable, consistently producing high levels of the recombinant protein across multiple attempts. The recombinant MLipA also exhibited increased activity compared to the unmodified LipA. Optimization studies revealed that a concentration of 0.2 mM IPTG, with an expression period of 24 h at 15°C, was adequate to induce strong expression of MLipA. Analysis conducted to determine the optimal IPTG concentration for MLipA expression showed minimal variation among the tested concentrations (Fig 2C), leading to the selection of 0.2 mM IPTG for subsequent recombinant MLipA expression.

Time course analysis (Fig 2D) demonstrated that an induction time of 24 h yielded the highest lipase activity, reaching 67.2 U/mL. The lipase activity at different IPTG concentrations was 8.4 U/mL (0 mM), 68.0 U/mL (0.2 mM), 63.1 U/mL (0.4 mM), 65.4 U/mL (0.6 mM), 62.4 U/mL (0.8 mM), and 60.9 U/mL (1.0 mM). For different incubation times, the lipase activity was 0 U/mL (0 h), 11.2 U/mL (4 h), 28.5 U/mL (8 h), 34.9 U/mL (12 h), 43.1 U/mL (16 h), 50.5 U/mL (20 h), 67.2 U/mL (24 h), 51.7 U/mL (28 h), and 39.3 U/mL (32 h). Beyond 28 h, both expression and activity of lipase declined. The aim of achieving expression of the recombinant lipase from *B. licheniformis* IBRL-CHS2 in a soluble bioactive form was successfully accomplished.

The purification of recombinant lipase MLipA was effectively carried out through immobilised metal affinity chromatography with TALON Metal Affinity Resin from Clontech, achieving in high specific activity of 331.3 U/mg (Fig 2E). The purification process, conducted in a one-step method and performed in triplicate, resulted in a final lipase recovery of 68% and a purification factor of 13 (Table 1).

**Purified MLipA activity screening on tributyrin substrate plate.** Following overnight incubation of filter paper discs treated with purified lipase on a tributyrin plate, clear hydrolysis zones were observed surrounding the three discs. Conversely, no such zones were seen around the negative control disc (Fig 3). These results demonstrate the ability of the purified lipase to effectively hydrolyze the substrate, confirming its full functionality.

## Characterization of recombinant MLipA

**Effect of temperature on MLipA activity and stability.** The enzyme experiment was run across a temperature range of 20°C to 45°C in order to identify the ideal temperature for MLipA activity (Fig 4A). According to a preliminary investigation, MLipA activity was first

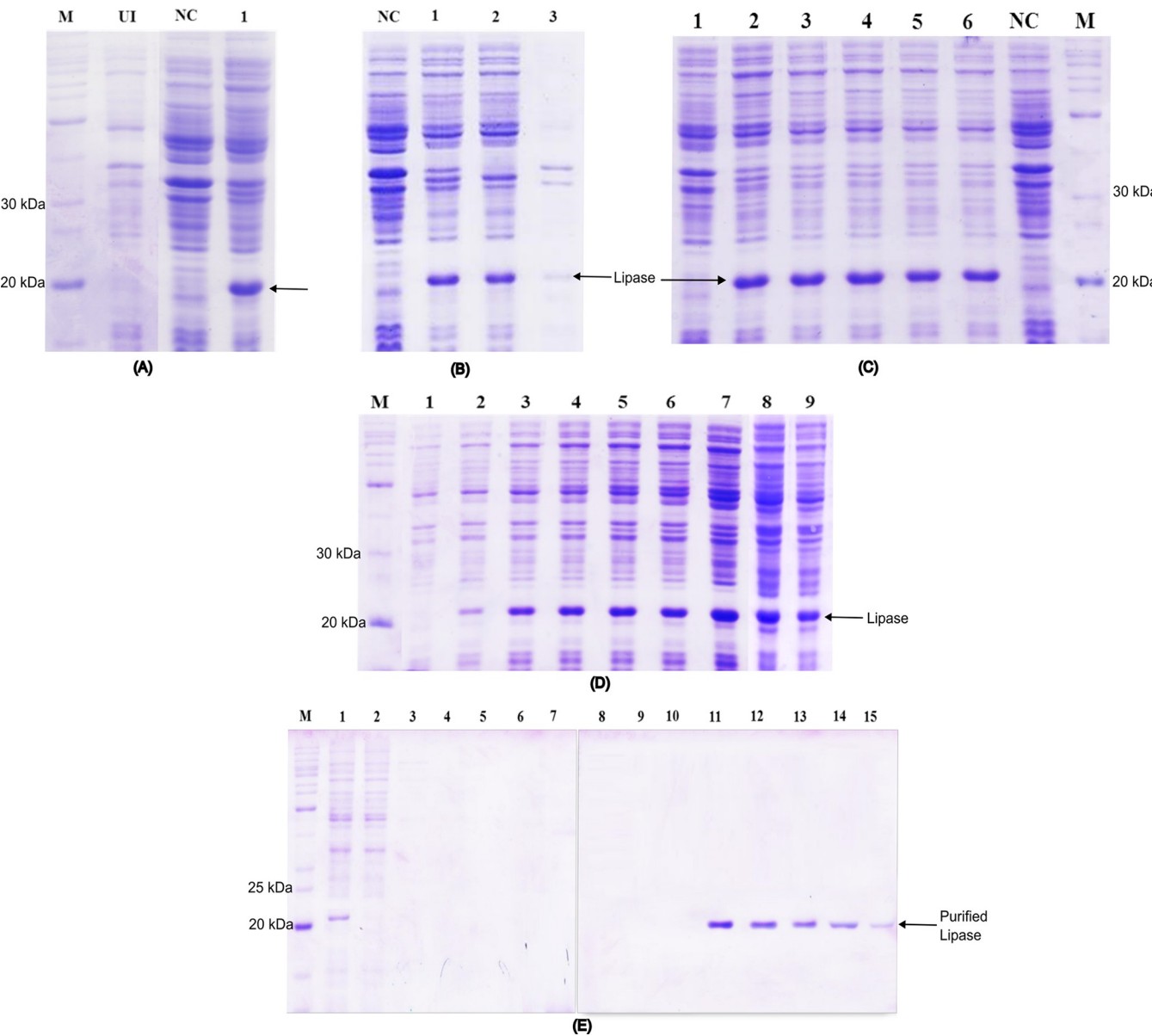

**Fig 2. Displayed all the result of SDS-PAGE analysis with Lane M as BenchMark^TM Protein Ladder and Lane NC is induced *E. coli* BL21 (DE3) harbouring empty pCold.** (A,B) MLipA$_{B.licheniformis}$ expression in *E. coli* BL21 (DE3) harbouring pCold-MLipA$_{B.licheniformis}$ (A) and solubility of the expressed lipase (B); Lane UI is uninduced *E. coli* BL21 (DE3) harbouring pCold- MLipA$_{B.licheniformis}$; Lane 1 is crude cell extract; Lane 2 is soluble proteins; Lane 3 is inclusion bodies. (C) Expression of MLipA$_{B.licheniformis}$ in *E. coli* BL21 (DE3) harbouring pCold-MLipA$_{B.licheniformis}$ after induction with different IPTG concentrations at 15°C. Lane1 is induction with 0 mM IPTG; Lane 2 is induction with 0.2 mM IPTG; Lane 3 is induction with 0.4 mM IPTG; Lane 4 is induction with 0.6 mM IPTG; Lane 5 is induction with 0.8 mM IPTG; Lane 6 is induction with 1.0 mM IPTG. (D) Time course analysis of the expression of MLipA$_{B.licheniformis}$ in *E. coli* BL21 (DE3) harbouring pCold- MLipA$_{B.licheniformis}$. Lane 1 is expression at 0 hour; Lane 2 is expression after 4 hours; Lane 3 is expression after 8 hours; Lane 4 is expression after 12 hours; Lane 5 is expression after 16 hours; Lane 6 is expression after 20 hours; Lane 7 is expression after 24 hours; Lane 8 is expression after 28 hours; Lane 9 is expression after 32 hours. (E) Purification of recombinant MLipA$_{B.licheniformis}$ by IMAC technique. Lane 1 is the diluted soluble crude lysate; Lane 2 is unbound proteins; Lanes 3–10 are wash fractions; Lanes 11–15 are eluate fractions containing purified lipase MLipA$_{B.licheniformis}$.

shown to rise gradually as the temperature rose near 35°C, until it peaked at 96% relative activity. After 40°C, the MLipA activity sharply decreased with additional temperature elevation.

Next, MLipA's thermostatability was investigated by first pre-incubating it for an h at various temperatures before doing a typical enzyme test. According to Fig 4B, MLipA was fairly

**Table 1. Purification table that summarizes the purification of MLipA via IMAC.**

| Purification step | Total activity (U) | Total protein (mg) | Specific activity (U/mg) | Purification fold | Yield (%) |
|---|---|---|---|---|---|
| Crude | 390 ± 4.70 | 15 ± 0.45 | 26 ± 0.90 | - | 100 ± 4.44 |
| IMAC | 265 ± 7.76 | 0.8 ± 0.01 | 331.3 ± 6.45 | 13 ± 1.90 | 68 ± 1.55 |

stable at 40°C with 83% of residual activity and maintained the majority of its enzymatic stability from 20°C to 35°C with over 95% of residual activity. But as the temperature rose over 40°C, there was a noticeable drop in MLipA's stability, and the enzyme lost up to 90% of its enzymatic activity. This demonstrated that MLipA exhibited thermolabile behavior, a characteristic shared by the majority of cold-adapted enzymes that are very susceptible to high temperatures. However, given that MLipA maintained 83% of its enzymatic activity at 40°C, it is possible that MLipA exhibits moderate thermolabile behavior.

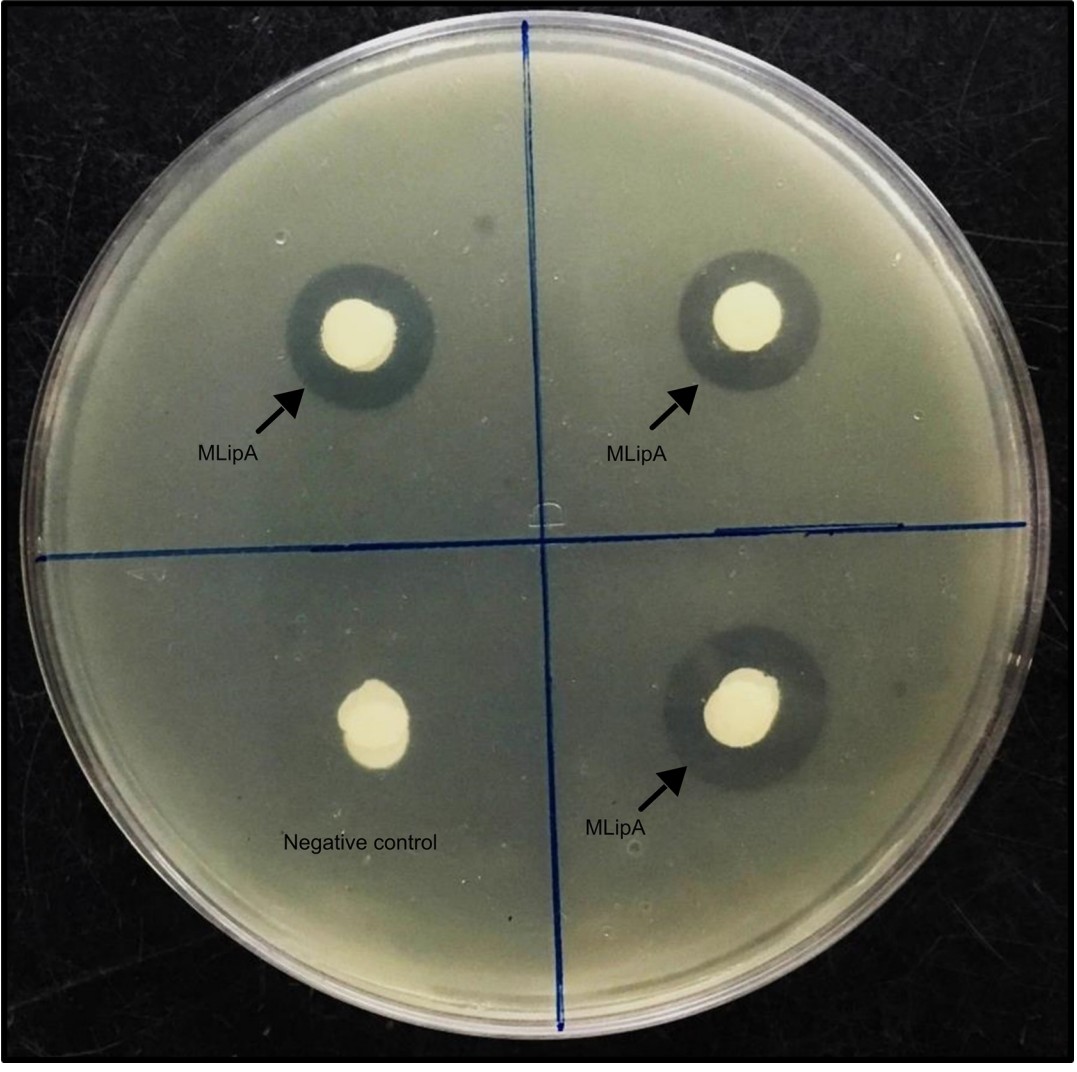

**Fig 3. Displays a tributyrin substrate plate indicating zones of hydrolysis caused by purified MLipA$_{B.licheniformis}$, as indicated by arrows in the image.** No clearance zone was observed around the negative control disc.

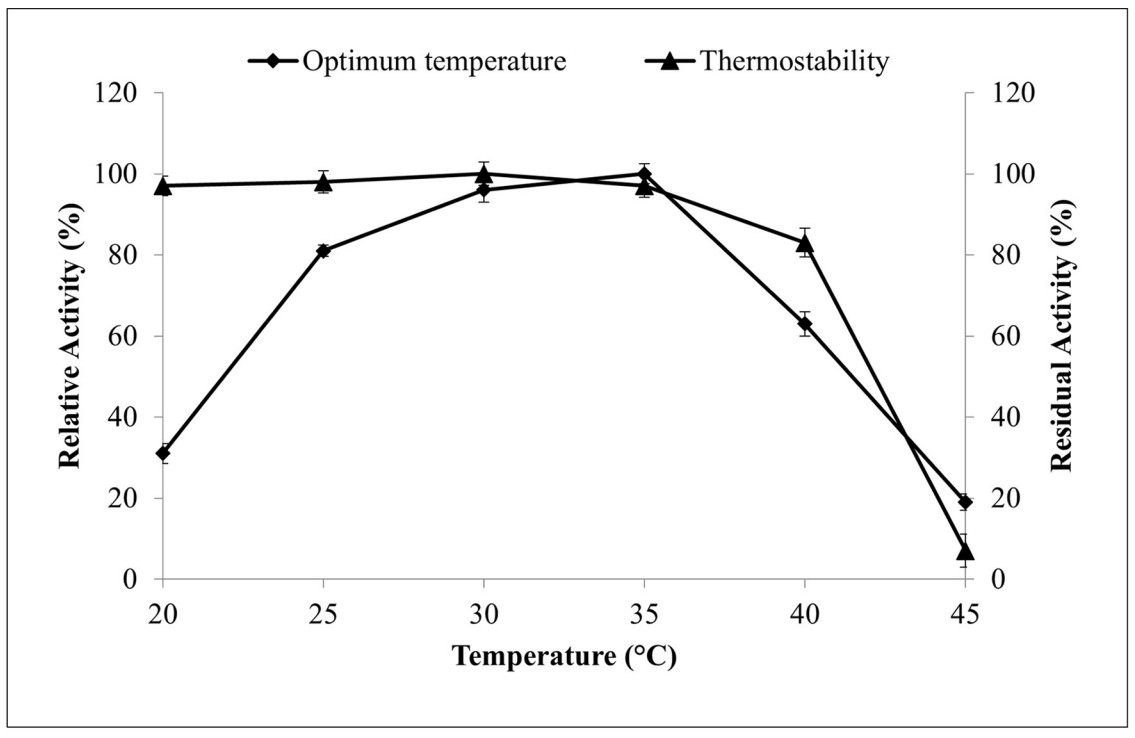

(A)

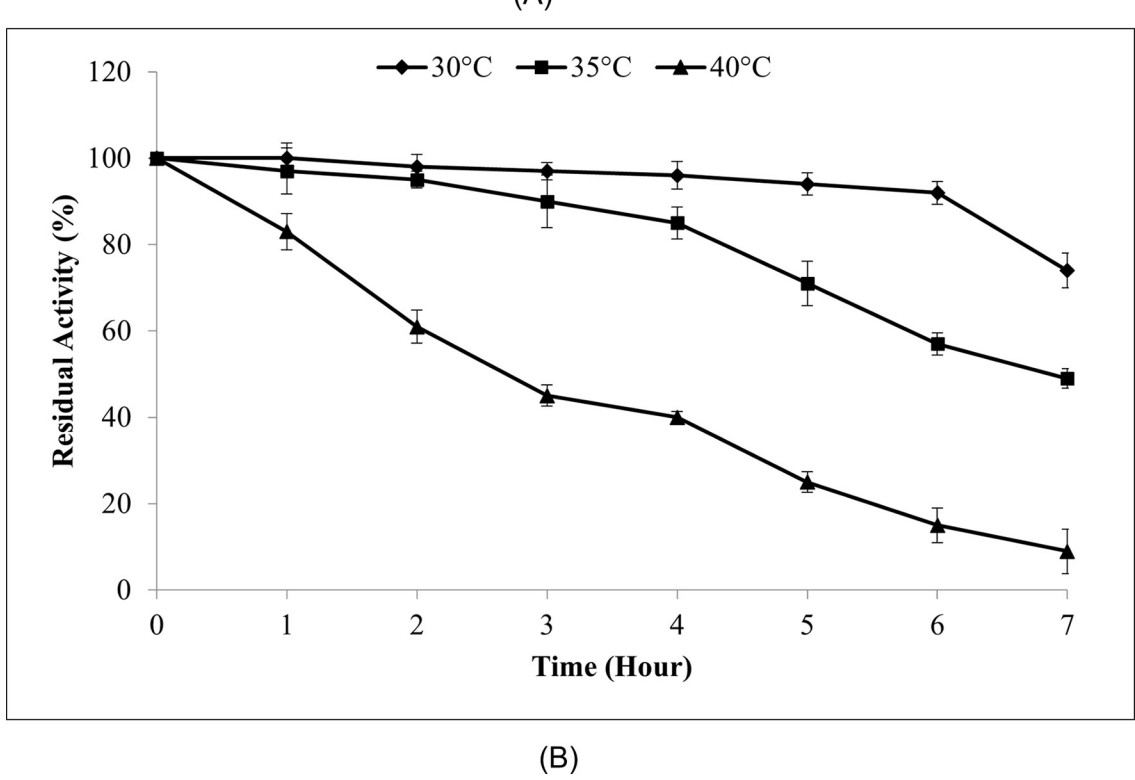

(B)

**Fig 4. Temperature profile and stability of MLipA$_{B.licheniformis}$.** (A) Temperature profile of MLipA$_{B.licheniformis}$ (♦) and temperature stability study (▲) of the lipase. To study the stability of lipase, the purified lipase was incubated in various temperatures for 1 hour prior to lipase assay. Residual activity was assayed at the optimum temperature, 35˚C. Specific lipase activity at 35˚C was set as 100% (361.6 U/mg) for determination of optimum temperature while for stability studies, the specific activity of lipase without pre-incubation (325.7 U/mg) was set as 100%. (B) Stability study of the lipase at 30˚C, 35˚C and 40˚C. To study the stability of lipase, the

purified lipase was incubated in the mentioned temperatures for 24 hours and aliquots were taken at specified intervals. Residual activity was assayed at the optimum temperature, 35˚C. Specific lipase activity without pre-incubation (325.7 U/mg) was set at 100%. Symbols used are: (♦) for 30˚C, (■) for 35˚C and (▲) for 40˚C.

Fig 4B shows the stability across three temperatures. It retained more than 90% activity after 16 h and 74% after 24 h, with a calculated half-life of 56 h under 30˚C. At 35˚C, over 90% activity remained after 2 h, declining to 49% after 24 h. However, at 40˚C, activity decreased significantly after 2 h (63%) and was just 9% after 24 h, indicating significant instability with increasing temperature. Despite this, MLipA exhibits moderate thermolabile behavior, notably retaining activity at 35˚C.

**Effect of pH on MLipA activity and stability.**   The recombinant lipase exhibited activity across a wide pH spectrum, ranging from pH 5 to pH 9, with optimal activity observed at pH 7 (Fig 5A). It maintained high activity levels, retaining 93% of its activity at pH 6 and 82% at pH 8. However, its relative activities decreased significantly at pH 4 (11%) and pH 5 (45%). Beyond pH 9, a sharp decline in activity was observed, with relative activities of 46%, 22%, and 8% at pH 9, pH 10, and pH 11, respectively.

The lipase's stability was assessed across various pH levels. Fig 5A illustrates that it remained stable between pH 6 and pH 9, retaining between 89% and 97% of its residual activity after 1 h of incubation at 35˚C. However, exposure to pH 10 resulted in a marked decrease in activity, with only 51% residual activity observed after one h, and further decline to 22% at pH 11. Subsequently, the stability of MLipA was evaluated across pH levels of 6, 7, and pH 8 (Fig 5B). It exhibited the highest stability at pH 6 and pH 7, retaining 43% and 49% of its original activity, respectively, following 24 h of incubation at 35˚C. Conversely, a substantial decrease in activity was observed at pH 8, with only 15% of the original activity remaining after 24 h.

**Effect of organic solvents on MLipA activity.**   MLipA stability in 10 different organic solvents with varying Log *P* values was evaluated over one and 24-h periods. Log *P* values indicate a solvent's hydrophilicity or hydrophobicity, with more positive values denoting greater hydrophobicity and more negative values indicating higher hydrophilicity. As shown in Table 2, the recombinant lipase from *Bacillus licheniformis* IBRL-CHS2 exhibited high tolerance to 25% (v/v) organic solvents. After 1 h, the lipase activity was notably enhanced in Isooctane (log *P* 4.5), DMSO (log *P* -1.22), n-hexane (log *P* 3.6), and methanol (log *P* -0.76), with relative activities of 192%, 161%, 159%, and 114%, respectively, compared to the control. Other solvents showed varying effects: n-heptane (log *P* 4.27) at 98%, ethanol (log *P* -0.24) at 97%, acetonitrile (log *P* -0.15) at 76%, and acetone (log *P* -0.23) at 61%. Diethyl ether (log *P* 0.85) retained 55% activity, while chloroform (log *P* 2.0) reduced activity to 28%. After 24 h, Isooctane, DMSO, n-hexane, and methanol continued to enhance activity, with relative activities of 181%, 139%, 114%, and 105%, respectively. Ethanol's effect slightly diminished to 74%. Other solvents further decreased lipase activity, resulting in 52% for acetonitrile, 38% for diethyl ether, and 83% for n-heptane. Only 4% relative activity remained with chloroform, while acetone retained 68% of activity.

**Substrate specificity of MLipA.**   The evaluation of the substrate specificity of the recombinant lipase from *Bacillus licheniformis* IBRL-CHS2 was conducted using different chain length *p*-nitrophenyl fatty acyl esters. This lipase showed a predilection for substrates that included medium- to long-chain fatty acids, as seen in Fig 6A. When *p*-nitrophenyl laurate (C12:0) was used as the substrate, the maximum activity was seen. *p*-nitrophenyl decanoate (C10:0) had the second-highest relative activity, at 89%, and *p*-nitrophenyl myristate (C14:0), at 71%. Relative activity of substrates like *p*-nitrophenyl palmitate (C16:0) and *p*-nitrophenyl octanoate

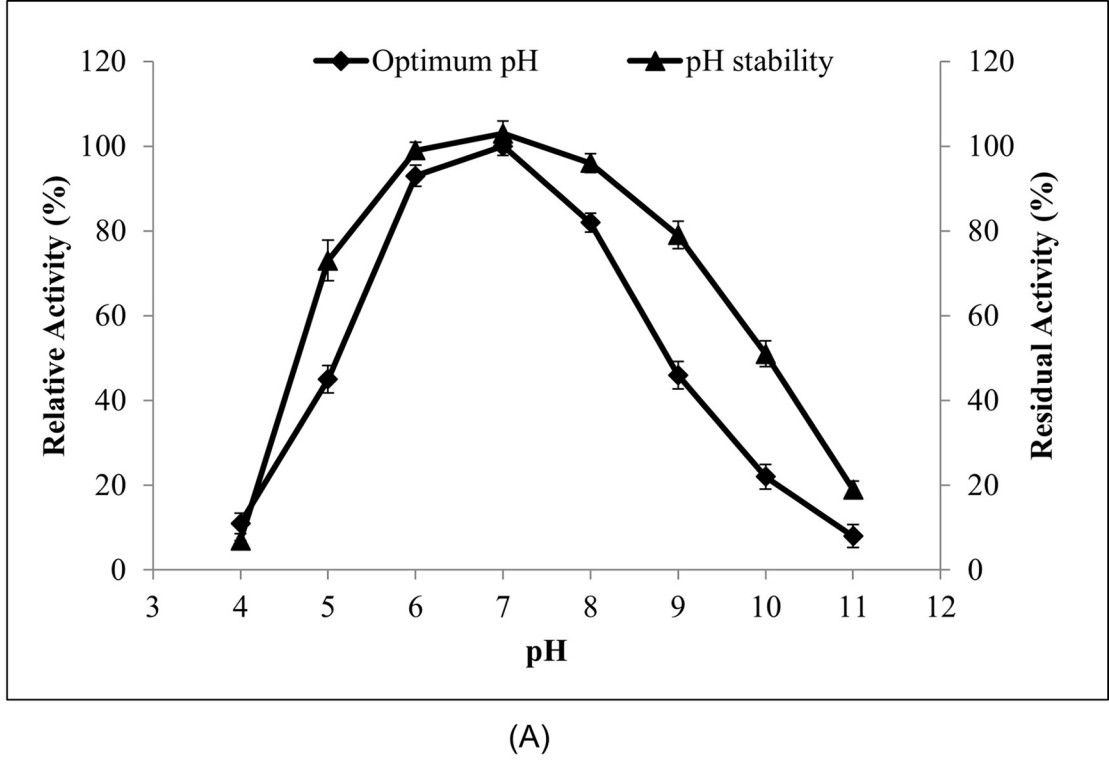

(A)

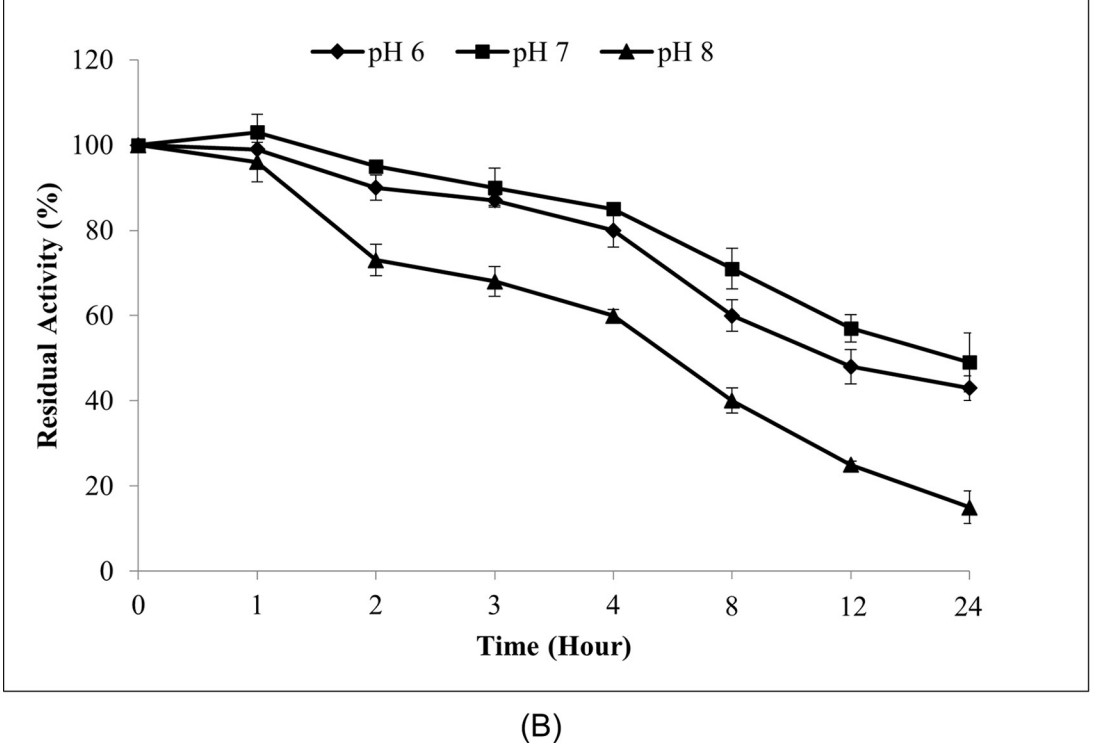

(B)

**Fig 5. pH profile and stability of MLipA_{B.licheniformis}.** (A) pH profile of MLipA_{B.licheniformis} (♦) and pH stability study of the lipase (▲). For stability study, the purified lipase was incubated for 1 hour in 35˚C in suitable buffers. Residual activity was assayed at the optimum temperature, 35˚C. Specific lipase activity at pH 7 was set as 100% (321.2 U/mg) for optimum pH while for pH stability, the activity of lipase without pre-incubation (325.7 U/mg) was set at 100%. (B) Stability study of MLipA_{B.licheniformis} at pH 6, pH 7 and pH 8. For stability study, the purified lipase was incubated for 24 hours in 35˚C in 0.1M phosphate buffer (pH 6 and pH 7) and

0.1M Tris-Cl buffer (pH 8). Aliquots were taken at specified intervals. Residual activity was assayed at the optimum temperature, 35°C. The specific activity of lipase without pre-incubation (325.7 U/mg) was set at 100%. Symbols used are: (♦) for pH 6, (■) for pH 7 and (▲) for pH 8.

(C8:0) were 40% and 44%, respectively. The substrate employed in this experiment, *p*-nitrophenyl stearate (C18:0), only showed 34% relative activity.

When tested against other natural oils, MLipA revealed a clear preference towards coconut oil exhibiting 56% relative activity over olive oil (shown in Fig 6B as 100%). It showed considerable activity toward palm oil (284% relative activity) and low activity toward canola and olive oils (79% relative activity). Lauric acid (C12), myristic acid (C14), and other medium chain fatty acids (MCFAs) including caprylic acid (C8) and capric acid (C10) make up the majority of the content of coconut oil, according to an analysis of the composition in S2 Table.

MLipA, on the other hand, is unable to effectively hydrolyze the other oils that were tested, which are mostly made up of long chain fatty acids including palmitic acid (C16), oleic acid (C18:1), linoleic acid (C18:2), and α-linoleic acid (C18:3). These findings are consistent with the substrate selectivity of MLipA that was previously addressed in relation to *p*-nitrophenyl esters.

**Determination of kinetic parameters of MLipA.** The analysis of kinetic parameters using *p*-nitrophenyl laurate as a substrate involved constructing a Lineweaver-Burk plot, as illustrated in S3 Fig. Based on the gathered experimental data, the $K_m$ value of the MLipA was determined to be 0.36 mM, with a corresponding $V_{max}$ of 357 µmol min$^{-1}$ mg$^{-1}$.

**Effect of metal ions on MLipA activity.** The impact of different monovalent, divalent, and trivalent metal ions on the activity of *Bacillus licheniformis* IBRL-CHS2 recombinant lipase was examined at concentrations of 1 mM and 5 mM as shawn in Fig 7. Results revealed that at a concentration of 1 mM, $Ca^{2+}$, $Na^+$, and $Ba^{2+}$ positively influenced lipase activity, with $Ca^{2+}$ showing the highest improvement (123%), followed by $Na^+$ (121%). Conversely, $Co^{2+}$ and $Cu^{2+}$ exhibited a slight enhancement in lipase activity, while $Mg^{2+}$ had no significant impact.

Upon increasing the metal ion concentration to 5 mM, it was observed that $Na^+$, $K^+$, and $Mg^{2+}$ had a positive effect on lipase activity, enhancing it. Conversely, an increase in the

**Table 2. Organic solvent stability study on MLipA in various hydrophobic and hydrophilic solvents.** Purfied lipase was incubated with 25% (v/v) organic solvent at a ratio of three volumes to one for durations of 1 h and 24 h at 35°C under 100 rpm. The activity was assayed after the incubation periods and relative activity was calculated against the control reaction (without addition of organic solvent) set as 100% (328.4 U/mg after 1 h incubation and 270.7 U/mg after 24 h incubation).

| Organic Solvents (25%) | Relative Activity (%) | | |
|---|---|---|---|
| | Log *P* | 1 h Incubation | 24 h Incubation |
| None | - | 100 ± 0.82 | 100 ± 1.44 |
| DMSO | -1.22 | 161 ± 2.52 | 139 ± 1.52 |
| Methanol | -0.76 | 114 ± 2.21 | 114 ± 1.21 |
| Acetone | -0.23 | 61 ± 0.90 | 68 ± 1.22 |
| Ethanol | -0.24 | 97 ± 1.10 | 74 ± 1.72 |
| Acetonitrile | -0.15 | 76 ± 1.20 | 52 ± 1.10 |
| Diethylether | 0.85 | 55 ± 3.2 | 38 ± 2.55 |
| Chloroform | 2 | 28 ± 0.95 | 4 ± 1.42 |
| *n*-hexane | 3.6 | 159 ± 1.71 | 105 ± 0.97 |
| *n*-heptane | 4.27 | 102 ± 3.56 | 83 ± 1.27 |
| Isooctane | 4.5 | 192 ± 2.08 | 181 ± 2.17 |

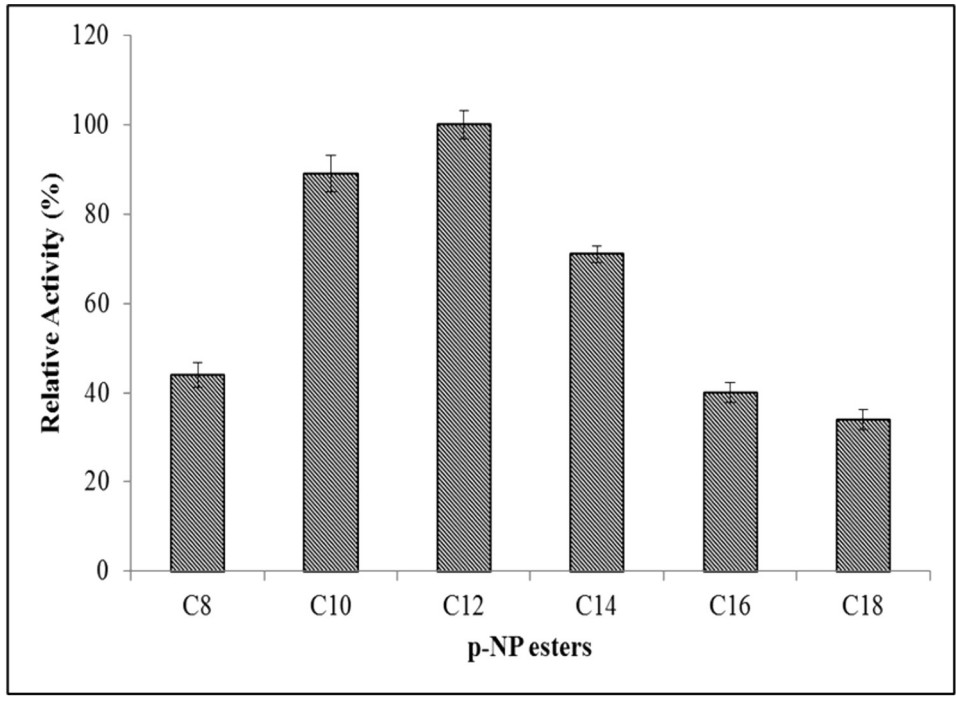

(A)

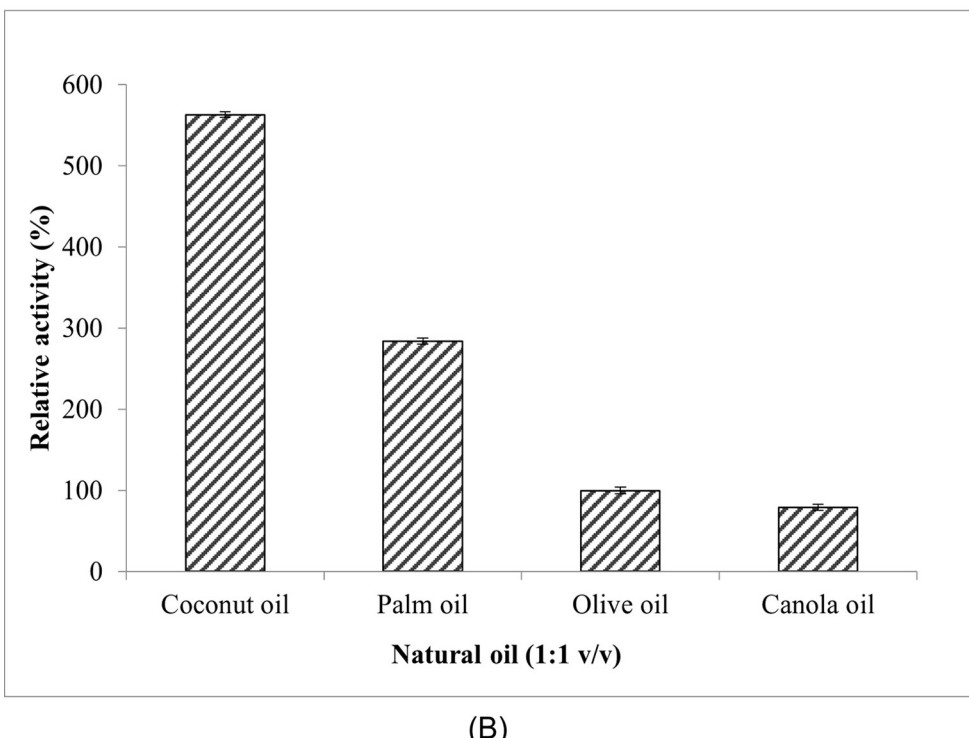

(B)

**Fig 6. Substrate specificity and Relative activity of MLipA$_{B.licheniformis}$.** (A) illustrates the MLipA$_{B.licheniformis}$ activity using various substrates. The experiments were carried out following standard assay protocols, with the substrate in the reaction replaced by different *p*-NP esters. The activity relative to lipase activity using *p*-NP laurate (C12) as the substrate was determined, with a value of 100% (318.5 U/mg). (B) displays the relative activity of MLipA$_{B.licheniformis}$ on different natural oils. The specific lipase activity of 36.6 U/mg when using olive oil as the substrate (control) was established as 100%.

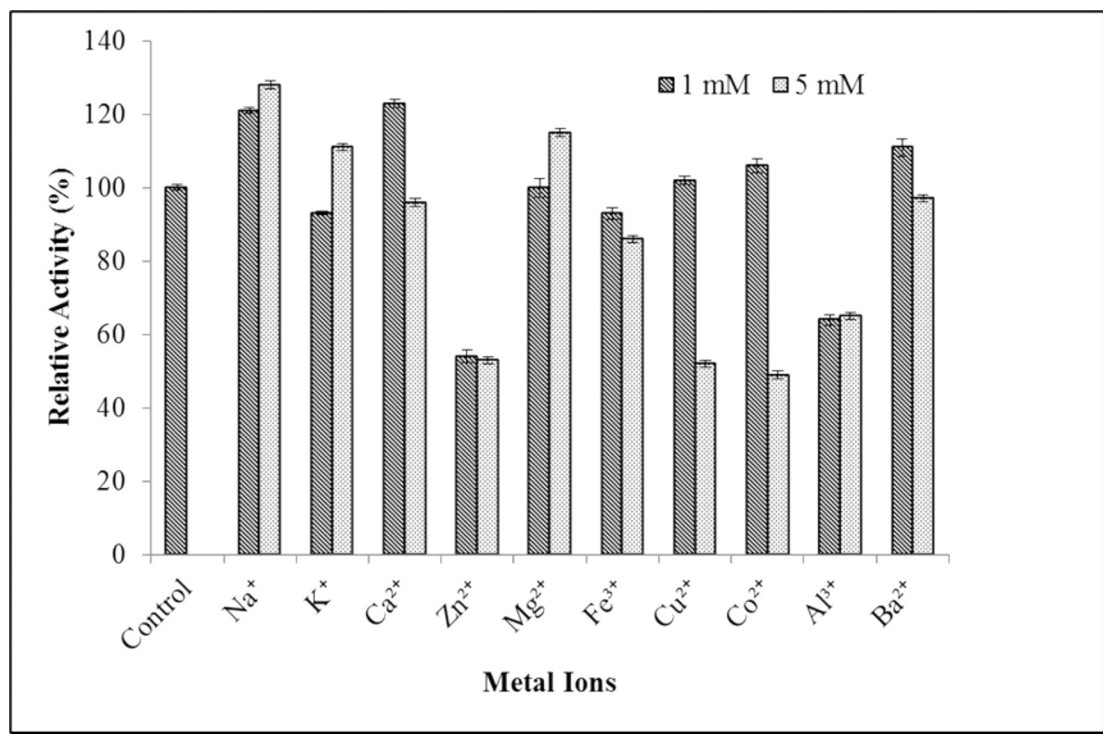

**Fig 7. The impact of metal ions on MLipA$_{B.licheniformis}$ activity.** The purified lipase was incubated with various additives for 1 hour at 35˚C and 100 rpm. The activity was assayed after the incubation and relative activity was calculated against the control reaction (without any additives) set as 100% (324.4 U/mg).

concentration of $Ca^{2+}$, $Ba^{2+}$, $Cu^{2+}$, $Co^{2+}$, and $Fe^{3+}$ had an inhibitory effect on lipase activity. Additionally, $Zn^{2+}$ and $Al^{3+}$ were found to decrease lipase activity at both concentrations.

**Effect of effector molecules on MLipA activity.** Fig 8 displays the findings of an investigation into the impact of several effectors on MLipA activity at 1 mM and 5 mM doses. Two oxidizing agents, two reducing agents, and two metal chelating agents were evaluated. Ammonium persulfate had no effect at 1 mM but increased activity by 17% at 5 mM. Potassium iodide enhanced activity by 6% at 1 mM but inhibited it by 13% at 5 mM. Ascorbic acid and 2-mercaptoethanol, two reducing agents, somewhat reduced lipase activity at both doses, which is unexpected since no cysteine residues are present in the lipase sequence to form disulfide bonds.

**Effect of surfactants on MLipA activity.** Fig 9 shows the effects of different surfactants on MLipA activity at a concentration of 1 mM. Triton X-100, a non-ionic detergent, notably enhanced MLipA activity by 207%. Similarly, other non-ionic detergents such as Nonidet P40, Tween 20, Tween 40, and Span 40 increased activity by 101%, 77%, 21%, and 5%, respectively. Considering the significant enhancement in lipase activity with just 1 mM of these surfactants, they offer a cost-effective means of enhancing MLipA performance as an industrial biocatalyst. However, slight inhibition was observed with Tween 80, and the anionic surfactant SDS strongly inhibited activity, reducing it to 51%.

## Discussion

### Sequence homology and phylogenetic analysis

The comparative study between lipase from *Bacillus licheniformis* (LipA) and other lipases from *Bacillus* provides detailed insights into its structural and functional characteristics where

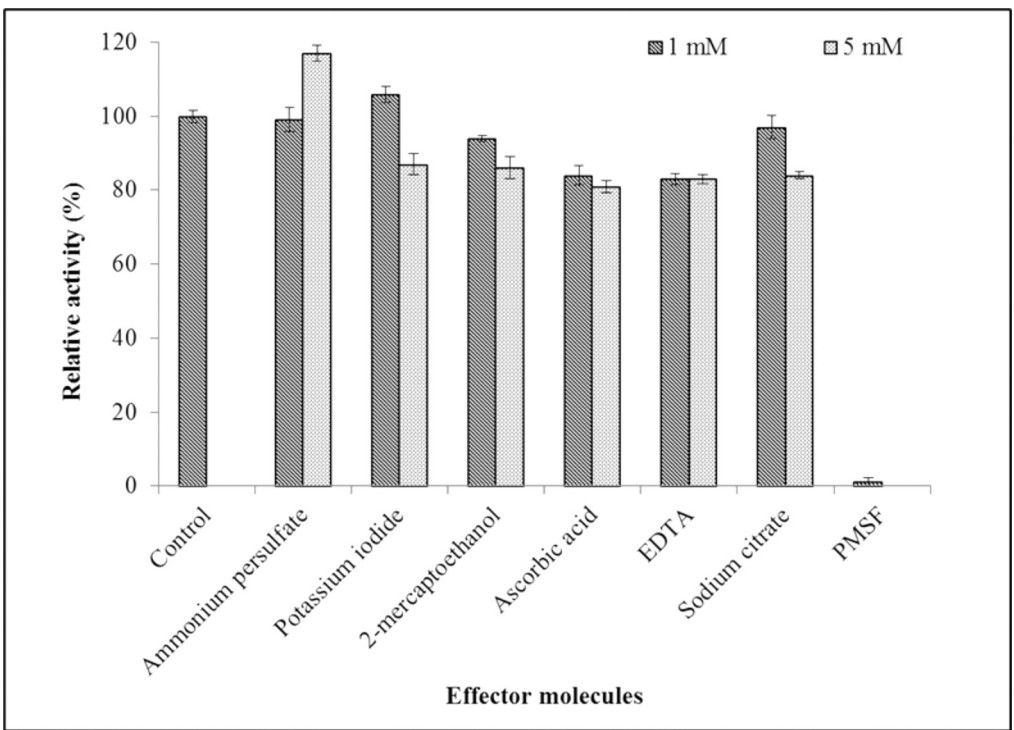

**Fig 8. The effect of effector molecules on MLipA$_{B.licheniformis}$ activity.** The purified lipase was incubated in the presence of the various additives for 1 hour at 35°C and 100 rpm. The activity was assayed after the incubation and relative activity was calculated against the control reaction (without any additives) set as 100% (328.3 U/mg).

it was found to align with the established catalytic triad observed in *B. subtilis* lipases [31]. These lipases are among the smallest known, as it has a molecular weight of around 19–20 kDa and share the conserved pentapeptide A–H–S–M–G [32]. Additionally, they are distinguished by their lidless feature, a trait shared with the lipase from *Bacillus licheniformis* IBRL-CHS2. Because the mature lipase sequence is 7 amino acid residues shorter than those of other sub-family 1.4 lipases, the mature lipase sequence, which consists of 174 amino acids, has a molecular weight of 18.5 kDa.

Using 1000 bootstraps and the Neighbour-Joining technique, a phylogenetic tree was built to investigate the evolutionary relationships among the lipases in subfamily 1.4 (Fig 1C). Of the eight lipases in subfamily 1.4 that have a common ancestor, LipA is the only one that is not clustered with the other *Bacillus* lipases and seems to be the most different. This divergence might be explained by the average 75% homology of amino acid sequences found in all examined *Bacillus* lipases, except for LipA [33,34]. Nonetheless, LipA and them share fewer than 70% (about 65–69%) of the same amino acids. The phylogenetic tree constructed for family 1 of bacterial lipases further supports this observation, showing that the LipA gene clusters with other subfamily 1.4 lipases, which are closely related to subfamilies 1.3 and 1.7, indicating a common ancestor (Fig 1D). The structural and molecular weight homology of *Bacillus* lipases with other members of family 1 of bacterial lipases is minimal [35].

## Expression and purification of recombinant MLipA

The expression of recombinant lipase from *B. licheniformis* IBRL-CHS2 was accomplished by employing the approach informed by the work of Gopal & Kumar [36], who demonstrated that excluding the signal peptide can improve both the expression and stability of recombinant

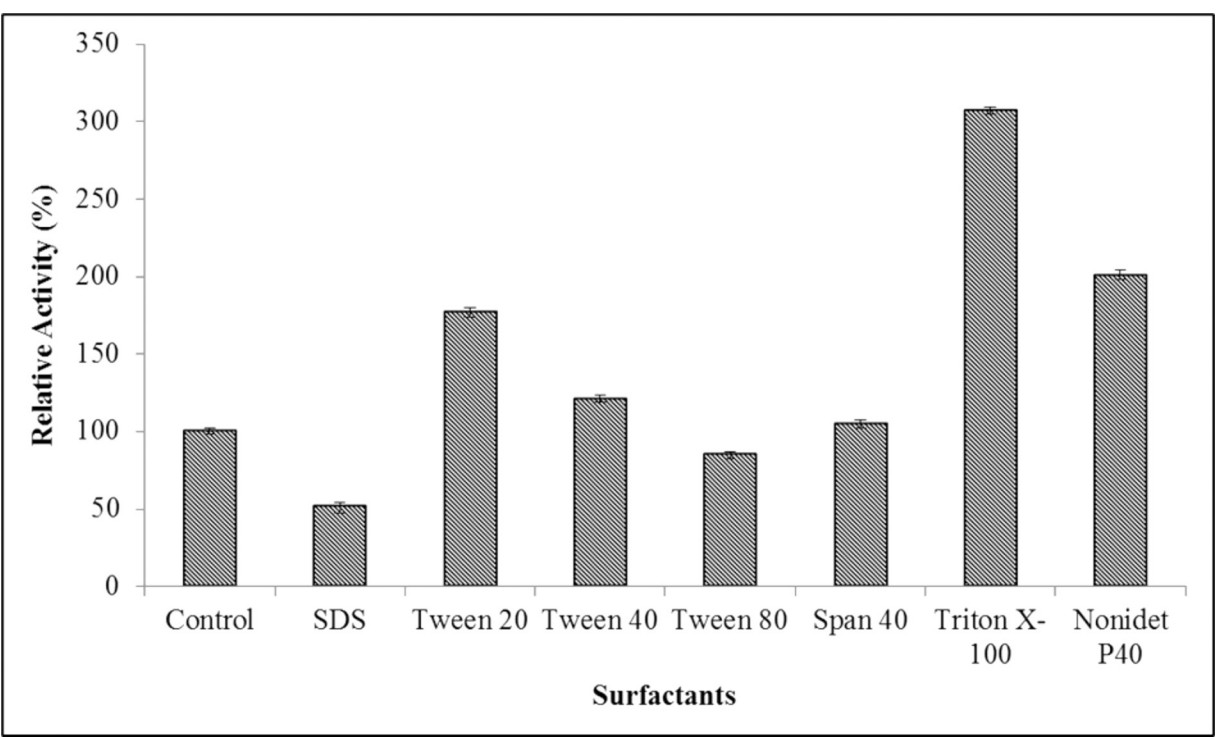

**Fig 9. The effect of surfactants on MLipA$_{B.licheniformis}$ activity.** The purified lipase was incubated in the presence of the various additives for 1 hour at 35°C and 100 rpm. The activity was assayed after the incubation and relative activity was calculated against the control reaction (without any additives) set as 100% (322.1).

proteins. Signal peptides are typically cleaved from endogenous proteins before they are exported to their target sites, and they do not contribute to the biological function of the protein. Therefore, removing the signal peptide from the open reading frame (ORF) is advisable if it hampers protein expression. Rahman, et al. (2005a) noted that signal peptides can promote the formation of inclusion bodies due to their high hydrophobic content, which may explain the aggregation of LipA expressed under the cspA promoter [37]. Consequently, the signal peptide coding region was removed from LipA, resulting in the successful expression of MLipA in *E. coli* BL21 (DE3) using the pCold I vector.

Several studies support the advantage of removing signal peptides for lipase expression. Jiang et al. (2010) reported a 4.3-fold increase in the expression level of a lipase from *Geobacillus stearothermophilus* JC after removing its signal peptide [12]. Similarly, Leow et al. (2007) found that eliminating the signal peptide from a thermo-alkaliphilic T1 lipase from *Geobacillus* sp. strain T1 increased its solubility and expression level by 3.8 times [38]. In another study conducted by Kim et al. (2000), the mature version of a thermostable lipase from *Bacillus stearothermophilus* demonstrated significantly higher activity without the signal peptide, whereas the precursor form primarily formed inclusion bodies, likely due to incomplete processing [39]. In summary, excising the signal peptide coding region from LipA significantly enhanced its expression and activity when utilizing the pCold I vector and *E. coli* BL21 (DE3) host system. This strategy of excluding the signal peptide has proven effective in enhancing the expression and functionality of lipases, making it a valuable approach for producing active recombinant enzymes.

Furthermore, as the pCold vectors require a temperature of 15°C for protein expression, no further investigation was required for the best post-induction incubation temperature. The

cold shock proteins produced at this temperature play a crucial role in initiating the expression of recombinant proteins. Besides, the utilization of *E. coli* as a host offers advantages such as cost-effectiveness and economical production of MLipA, as it eliminates the need for expensive chemicals or fusion tags. A low induction concentration (0.2 mM IPTG) is adequate to initiate lipase synthesis, eliminating the need for additional downstream processes.

The effectiveness of the one-step purification method using immobilised metal affinity chromatography with TALON Metal Affinity Resin was proven, as illustrated in Table 1. Comparatively, other lipases purified using similar His-tag affinity purification techniques exhibited different varying degrees of recovery and purification factors. For instance, the lipase derived from *Geobacillus* sp. strain ARM demonstrated a recovery of 63.2% and a purification factor of 14.6 [40], while the lipase of *Pseudomonas aeruginosa* CS-2 exhibited a recovery of 40.9% and a purification factor of 10.2 [41]. The *Serratia marcescens* ECU1010 lipase achieved a 2.4-fold purification with a recovery rate of 59.4% [42]. Additionally, the *B. subtilis* DR8806 lipase had a 57% yield and purification factor of 3 [43].

In contrast, the purification of the recombinant lipase from *B. amyloliquefaciens* PS35 required a two-step process involving Phenyl Sepharose hydrophobic interaction chromatography followed by ion exchange chromatography. This method yielded a final recovery rate of 9.7% [44]. Ma et al. utilized a three-step purification technique for a recombinant *B. subtilis* lipase, achieving a 19.7-fold purification with a final yield of 30% [45]. Among these methods, the purification of MLipA with a 68% recovery stands out for its efficiency in terms of time and cost savings associated with the use of the His-tag.

The estimated molecular weight of MLipA is 18.5 kDa, slightly smaller than other lipases in the same genus, which typically range from 19–20 kDa [32]. The addition of a 3 kDa extension at the N-terminal, part of the pCold I vector, resulted in the purified MLipA band to migrate as a single homogeneous band of approximately 21.5 kDa. Comparatively, the cloned mature LipA of *Bacillus licheniformis* and lipases from *Bacillus pumilus* B26, *Bacillus subtilis*, and *B. amyloliquefaciens* have similar molecular weights around 19 kDa as reported in previous studies [18,19,33,46,47].

## Characterization of recombinant MlipA$_{B.\ licheniformis}$

**Effect of temperature on MLipA activity and stability.** Enzymatic catalysis emphasizes that subfamily I.4 lipases are characterized as mesophilic enzymes, exhibiting an optimal temperature usually around 35˚C. The absence of disulfide bonds, crucial for stabilizing protein structures, in Subfamily 1.4 lipases likely accounts for their diminished stability at elevated temperatures. Consequently, researchers have explored strategies such as introducing disulfide bonds into lipases to enhance both their catalytic activity and thermostability [48].

Conducting industrial operations at moderate temperatures serves to suppress any undesirable side reactions that may occur at elevated temperatures, a particularly crucial consideration in the industries like pharmaceutical, fine chemical synthesis, or food processing. Within these sectors, the preservation of heat-labile substrates and products is paramount [49,50]. Specifically, in the realm of food processing, maintaining reactions at low to moderate temperatures is imperative to preserve volatile molecules and prevent alterations in flavor and other constituents, thereby safeguarding the nutritional value and taste integrity of the final products [50]. An additional advantage of heat-sensitive MLipA is its susceptibility to inactivation through a simple elevation of the reaction temperature to 45˚C, obviating the need for harsh chemical treatments.

**Effect of pH on MLipA activity and stability.** The activity of enzymes is greatly affected by pH. Researchers have extensively examined how pH impacts both the stability and activity

of enzymes across various types and applications. Tavares et al. underscored the influence of low pH on enzyme structure stability [51]. Balakrishnan et al. further examined this phenomenon by exploring how varying pH levels during the incubation of crude enzymes affect their stability [52]. El-Ghonemy emphasized the substantial influence of pH changes on the ionic properties of enzyme groups, consequently impacting enzyme conformation, shedding light on the pivotal role pH plays in modulating enzyme functionality [53].

Fashogbon et al.revealed that bacterial lipases typically exhibit optimal activity at neutral or alkaline pH levels, although exceptions exist, such as the lipase from *P. fluorescens* SIK W1, which displays peak functionality at an acidic pH of 4.8 [54]. This diversity suggests that lipase activity preferences may vary among bacterial species. O A et al. investigated lipases produced by *Brevibacterium*, *Bacillus*, and *Pseudomonas* spp., emphasizing the criticality of identifying optimal conditions, including pH, for efficient lipase production [55]. Their findings underscore the importance of elucidating the pH requirements of lipases derived from distinct bacterial genera, contributing to a deeper understanding of enzymatic activity under varying environmental conditions.

**Effect of organic solvents on MLipA activity.** The influence of organic solvents on lipase activity is a crucial area of research that has attracted significant attention due to its implications for various biocatalytic processes. Organic solvents play a key role in modulating the catalytic performance and stability of enzymes, such as lipases, in different systems. Studies have demonstrated that the physicochemical properties of organic solvents, including their chemical functionalities and hydrophobicity, are fundamental determinants of enzymatic activity and robustness in biphasic systems [56].

One explanation for the enhanced stability and activity of MLipA in DMSO, ethanol, and methanol could be the diffusion of these solvents into the enzyme's active site, replacing water molecules and increasing flexibility, as suggested by Jóhannesson et al. in their study on lysozyme [57]. They noted that DMSO can act as a stabilizer, denaturant, inhibitor, activator, or cryoprotectant, depending on the interactions within the DMSO-protein complex. Lesuisse et al. also attributed DMSO's stimulatory effect to its capability to enhance substrate concentration available to *B. pumilus* 168 lipase and inhibit enzyme aggregation [58]. More recently, (Dachuri et al. found that cold-adapted lipases PML and LipS underwent conformational changes in the presence of DMSO, methanol, and ethanol, resulting in increased stability and solvent tolerance [59]. Therefore, the stability and enhanced activity of MLipA in these solvents likely stem from these combined factors. Given MLipA's stability in methanol and ethanol, commonly used in biodiesel production, it has potential as a biocatalyst for producing fatty acid methyl esters (FAME), potentially reducing the costly production associated with chemical catalysts [60].

In isooctane and n-hexane, MLipA exhibited enhanced activity while remaining stable in n-heptane. Hexane is the recommended solvent for the majority of transesterification processes in ester synthesis, according to Gricajeva, et al. [35]. Similar to this, *Bacillus* sp. lipase showed increased activity in n-hexane and decreased activity in chloroform [61]. According to Liu et al., the *B. subtilis* DS9 lipase also showed notable increases in activity in 25% (v/v) n-hexane (258% relative activity), heptane (256%), and isooctane (321%), but it was inhibited in methanol and ethanol [62]. The capacity of enzymes to withstand denaturation by creating numerous hydrogen bonds with water molecules, maintaining structural flexibility and catalytic activity, is thought to be responsible for their increased stability and activity in organic solvents [63]. Furthermore, it has been shown by Sharma & Kanwar that enzymatic catalysis is further promoted by forcefully mixing aqueous and organic solvents since this provides a wide interfacial area [64].

MLipA exhibited superior stability in organic solvents compared to the *B. licheniformis* RSP-09 lipase. After exposure to 25% (v/v) of various organic solvents for 10 h, the RSP-09 lipase showed different levels of activity retention: 54% in isooctane, 65% in n-hexane, 96% in DMSO, and 23% in acetonitrile [18]. The inhibitory effects observed with choloroform, diethyl ether, and acetone on MLipA may be due to these solvents altering the aqueous environment around the lipase, affecting the buffer's dielectric constant and penetrating the enzyme's core, thereby leading to inactivation [65]. Immobilizing MLipA could further enhance its stability in organic solvents, enabling repeated use over extended periods [66].

It has been investigated how plant lipase and organic solvents are used as reaction media in plant oil lipolysis, highlighting the significance of solvent properties such as dielectric constant and viscosity in influencing lipase performance during lipolysis. Lipases have been studied in dry organic solvents to assess their activity as a criterion, demonstrating the versatility of these enzymes in various solvent environments [67]. The presence of non-polar organic solvents has been associated with improved thermostability of lipases, suggesting the potential for enhanced enzyme performance in such environments [68]. Studies have also emphasized the significance of organic solvent properties, such as dielectric constant and viscosity, in impacting lipase performance during lipolysis [69].

Furthermore, previous research demonstrated that the addition of organic solvents significantly alters enzyme activity and stability, with some solvents enhancing enzyme performance while others may lead to inactivation [70]. The interaction between enzymes and organic solvents can have diverse effects, with certain solvents improving enzyme conformation and activity, while others may denature the protein structure, leading to decreased or even loss of activity [71]. Organic solvents have the ability to remove water molecules from the surface of an enzyme, allowing them to enter the active site further and denaturing proteins [72].

**Substrate specificity of MLipA.** A significant difference between lipases and esterases lies in the capacity of lipases for the hydrolysis of longer chain fatty acid esters, whereas esterases are limited to hydrolyzing short chain fatty acid esters [73,74]. This difference can be explained by the fact that lipases have a bigger, hydrophobic scissile fatty acid binding pocket than esterases, which have a smaller acyl binding site. The shape and size of lipases' active sites are critical factors in defining their substrate selectivity [74,75].

Study by Jeon et al. demonstrated the recombinant EML1 lipase from deep-sea sediment metagenome exhibited a preference for coconut oil, with subsequent preferences for palm oil and olive oil [76]. Similar preferences were shown by the lipase from *Bacillus* sp. RN2 for coconut oil, olive oil, and palm oil. The lipase derived from *Pseudomonas* sp. strain S5, on the other hand, showed a stronger predilection for palm oil than for coconut or olive oil. Leow et al. reported inconsistent findings, indicating that the *Geobacillus* sp. T1 lipase exhibited increased activity towards palm and olive oils and decreased activity towards coconut oil [38]. Furthermore, compared to coconut oil, the lipase of *Bacillus* sp. strain L2 showed a greater affinity for olive oil, suggesting that lipases have varying preferences for different types of oil.

The production of fatty acids from oils is commonly utilized in industries such as oleochemical, cosmetic, food, and pharmaceutical [77,78]. Currently, the preferred method for fatty acid production is through lipases due to their milder reaction conditions and energy efficiency compared to chemical and physical means [79–81]. Given MLipA's high affinity for the C12 substrate, it could potentially aid in producing lauric acid from coconut oil and palm kernel oil, which are abundant sources of this fatty acid (comprising 51.6% of C12:0). Lauric acid is known to be an effective antimicrobial agent in both cosmetic and food industries [77,82,83]. Furthermore, coconut oil may be processed with lipase to create additional MCFAs like lauric acid, which adds value to the product and makes it suitable for therapeutic use.

Thus, these oils may be used to generate lauric acid and other MCFAs that are advantageous for the food, cosmetic, and pharmaceutical sectors by employing MLipA as a biocatalyst.

The fact that the recombinant MLipA able to hydrolyze long chain substrates and prefers medium chain substrates suggests that it is a genuine lipase. Consistent with these results, the lipase from *Geobacillus zalihae* T1 and another lipase from *Bacillus licheniformis* NCU CS-5 also exhibited a preference for *p*-nitrophenyl laurate [84,85]. In contrast, the lipase from *Geobacillus thermocatenulatus* demonstrated maximum preference for *p*-nitrophenyl myristate (C14:0), followed by *p*-nitrophenyl palmitate (C16:0) and *p*-nitrophenyl laurate [86]. On the other hand, LipA from *Bacillus licheniformis* and the lipase from *Geobacillus kaustophilus* DSM 7263T appeared to prefer short chain fatty acid esters.

**Determination of kinetic parameters of MLipA.** The Lineweaver-Burk plot has been widely recognized as a reliable method for elucidating the kinetic characteristics of various lipases, as evidenced by studies in the literature [35,44,87–89]. The substrate concentration at which half of an enzyme's active sites are occupied is represented by the Michaelis constant, or $K_m$, which reflects the enzyme's affinity for the substrate. With a lower $K_m$ value, the binding affinity is greater [90]. Typically, lipases exhibit Michaelis-Menten kinetics, with $K_m$ values falling in the range of between $10^{-1}$ to $10^{-5}$ M (0.1 to 0.00001 M) for industrial applications [91].

The $K_m$ value of MLipA acquired falls within the acceptable range for industrial use and is notably lower than the $K_m$ values reported for lipase from *B. amyloliquefaciens* (4.345 mM) and LipA from *B. licheniformis* (0.45 mM), both using *p*-nitrophenyl palmitate as the substrate [18,44]. In contrast, Ma et al. found that the utilisation of *p*-nitrophenyl octanoate on *B. subtilis* lipase as the subsrtrate showed $K_m$ values of 0.37 mM and 303 μmol min$^{-1}$ mg$^{-1}$ for $V_{max}$ [45], which are comparable to those of MLipA [89] suggested that absence of a lid domain in subfamily 1.4 lipases enables them to operate effectively at lower substrate concentrations and exhibit enhanced substrate affinity.

**Effect of metal ions on MLipA activity.** Metal ions function as electrophiles and have a propensity to share electrons with other atoms, which can lead to the creation of bonds or charge-charge interactions between them, according to Glusker et al. [92]. Enzyme activity may be increased or decreased as a result of this interaction. Additionally, it has been proposed that some metal ions may form bonds with the ionic side chains of particular amino acids, serving as cofactors in catalytic activity, whereas other metal ions may break these bonds which lead to denaturation at the active site (non-competitive inhibition). By stabilizing or destabilizing an enzyme's structure, metal ions can also affect the activity of the enzyme [93,94].

Upon evaluation of the impact of metal ions on recombinant lipase MLipA, it was observed that all bivalents tested, with the exception of $Zn^{2+}$, enhanced the lipase activity at a concentration of 1 mM. Among these bivalents, $Ca^{2+}$ exhibited the greatest enhancement in MLipA lipase activity, consistent with previous findings indicating a 10–21% increase in activity with the presence of $Ca^{2+}$ [95]. This enhancement can be attributed to the stabilizing effect of $Ca^{2+}$ on the enzyme structure, which helps maintain catalytic residues in their proper position and bridges the active site region to a secondary subdomain, resulting in tertiary structure stabilization [38,39,96].

According to a 1993 study by Lesuisse et al., metal ions may affect the catalytic mechanism of lipase by creating complexes with ionized fatty acids that alter the lipase's solubility and behavior at the interface [58]. $Ca^{2+}$ can increase lipase activity by forming insoluble Ca-salts of fatty acids that block product inhibition [97,98]. Numerous investigations have demonstrated that the presence of $Ca^{2+}$ enhanced the activity of lipases from *Bacillus licheniformis* NCU CS-

5 [99], *B. licheniformis* SCD11501 [100], *B. licheniformis* MTCC6824 [101], *B. cereus* C7 [102], *Bacillus* sp. L2 [103], and *B. subtilis* [45].

However, research by Emtenani et al. and Cai et al. found that $Ca^{2+}$ greatly reduced the enzyme activity of recombinant *B. subtilis* DR8805 lipases and *B. amyloliquefaciens* lipases [33,43]. In addition to $Ca^{2+}$, $Ba^{2+}$ showed only a slight increase in activity of lipase, while $Cu^{2+}$ and $Co^{2+}$ had minimal effect at 1 mM. However, at a higher concentration of 5 mM, both $Cu^{2+}$ and $Co^{2+}$ caused a significant inhibition of lipase activity, while $Ca^{2+}$ and $Ba^{2+}$ only had a slight inhibitory effect. However, when the concentration of the metal ion rose, the presence of $Mg^{2+}$ led to an increase in lipase activity, which is consistent with the results of Olusesan et al., who observed that $Mg^{2+}$ marginally stimulated *B. subtilis* NS-8 lipase [104]. The lipase activity of MLipA was negatively impacted by both amounts of $Fe^{3}$, and the levels of lipase activity were also dramatically reduced by $Zn^{2+}$ and $Al^{3+}$.

Dong et al. noted that transition metal ions including $Zn^{2+}$, $Cu^{2}$, $Fe^{3+}$, and $Co^{2+}$ may interact with an enzyme's surface charge to modify the conformation of the enzyme and change the ionization state of certain amino acids [105]. This alteration may cause the enzyme to become unstable, which might alter the activity of the enzyme and cause ion toxicity. According to experimental results, free $Zn^{2+}$ may negatively affect enzyme activity by causing conformational disruption and decreasing stability [29,38,94]. On the other hand, lipase stability and activation depend heavily on firmly bound $Zn^{2+}$ [106].

On the other hand, monovalent ions such as $Na^+$ and $K^+$ were found to enhance the activity of MLipA, particularly $Na^+$, similarly to the positive effect of $Ca^{2+}$ on the lipase. This aligns with previous research showing the stimulating effect of $Na^+$ and $K^+$ on lipase from *B. amyloliquefaciens* [44]. Additionally, $Al^{3+}$ was shown to significantly inhibit lipase activity. By contrast, Nthangeni et al. found that $Ba^{2+}$, $Cu^{2+}$, $Fe^{2+}$, $Zn^{2+}$, and $Co^{2+}$ inhibited LipA from *Bacillus licheniformis* rather than activating it [19].

**Effect of effector molecules on MLipA activity.** Effector molecules play a crucial role in modulating the activity of lipases. They can influence the catalytic activity of lipases by affecting their diffusional patterns, binding efficiency, and functional states [107]. These molecules can regulate enzymatic reactions by influencing the rate of catalysis [108].

Both *B. subtilis* DR8806 and *B. stratosphericus* L1 recombinant lipases lost activity in the presence of reducing agents despite lacking cysteine residues [35,43]. Metal chelating agents EDTA and sodium citrate slightly reduced lipase activity, indicating the enzyme requires metal ions for optimal activity and stability [95]. This reduction may also result from the chelating agents' negative impact on the interfacial area between substrates and the lipase [109]. Similar results were seen by Gang et al.for a lipase from *B. cereus* BF-3, which exhibited a modest decrease in activity when exposed to EDTA [110]. In contrast, the ARM lipase in *Geobacillus* sp. showed improved activity with EDTA, indicating that it is not a metalloenzyme [40].

Effector molecules can also influence the activity of lipases in different environments. For example, the presence of nonpolar compounds surrounding lipase molecules can potentially increase their activity in polar organic solvents [111]. Additionally, important variables influencing the catalytic activity of lipases include the binding of organic molecules into the active region of the enzymes and the substrate's water accessibility [112].

**Effect of surfactants on MLipA activity.** By boosting substrate solubility, maintaining enzyme conformation, averting enzyme aggregation, and enlarging the water-lipid interface, surfactants improve lipase activity [32]. According to Peng et al., they are often employed in lipase-catalyzed processes [113].

The impact of surfactants on lipase activity is a multifaceted process influenced by the type of surfactant used. Research has demonstrated that surfactants from various categories—cationic, anionic, and non-ionic—can have a significant effect on lipase activity [114]. Non-ionic

surfactants like Triton X-100 have been found to induce lipase activity, thereby reducing enzyme denaturation, while others like Tween and NP-40, as well as ionic surfactants such as SDS and CTAB, generally inhibit lipase activity [115,116]. Non-ionic surfactants have been identified as crucial enhancers of enzyme activity at cationic reverse micellar interfaces, potentially by modifying enzyme hydrophobicity [117]. Additionally, the addition of non-ionic surfactants has been shown to enhance lipase activity by reducing hydrophobic and electrostatic interactions between surfactants and lipases [118]. The concentration of surfactants in the reaction environment is another critical factor affecting lipase activity. Studies have highlighted that surfactant concentration is a key determinant in controlling the activity of lipases [119]. Moreover, surfactants can regulate interfacial catalysis by influencing lipase adsorption and mobility at hydrophobic surfaces [120].

SDS has been shown to strongly inhibit the activity of lipases. For example, in a study by Bora & Bora, SDS was found to be a strong inhibitor of *Bacillus thermoleovorans* lipases, causing almost total inhibition of enzyme activity [121]. Sukohidayat et al. reported that lipase activity was inhibited in the presence of 0.1% SDS, with lipase activity dropping to 21.36% after 30 min of incubation [115]. Additionally, Mohaini et al.found that SDS had a significant inhibitory effect on lipase activity, potentially causing local conformation changes in the enzyme's active region, leading to inhibition, partial reversible unfolding, and subsequent deactivation [122]. According to Peng et al., Tween 80 most likely caused a minor reduction in lipase activity by competitive interaction with substrate at the enzyme's active region or by significantly denaturing proteins [41,123]. *Pseudomonas* sp. DMVR46 lipase activity was shown to be somewhat reduced in the presence of Tween 80 and to be more strongly inhibited in the presence of SDS, but Triton X-100 was found to enhance the enzyme's activity [124].

## Conclusions

In conclusion, the lipase gene of the *Bacillus licheniformis* IBRL-CHS2 was cloned and identified as part of the subfamily 1.4 lipases, which are known for their small size, lack of a lid structure, and distinctive pentapeptide sequence A-H-S-M-G. Initially, the recombinant lipase expression was unsatisfactory. However, the production of mature lipase (MLipA) in *E. coli* BL21 (DE3) was significantly enhanced by removing the signal peptide and employing the pCold expression vector. The MLipA was then efficiently purified in one step using immobilized metal affinity purification. Characterization revealed that MLipA lacks thermostability and alkaline stability, with a preference for medium-chained esters over short and long-chained ones. The lipase demonstrated stability in the presence of most tested additives and surfactants, and it remained stable in various organic solvents. While many lipases lose activity in non-aqueous media, MLipA naturally remains stable and even shows enhanced activity in DMSO, methanol, n-hexane, and isooctane. It performs well at ambient temperatures, making it suitable for industrial applications such as pharmaceutical production, fine chemical synthesis, and food industry flavor modifications. MLipA does not require cofactors to function and works optimally with very low substrate amounts due to its lidless structure. Moreover, MLipA is an excellent candidate for protein engineering studies due to its unique characteristics of small size and efficient production in *E. coli*, facilitate genetic manipulation.

This research has identified a promising biocatalyst with potential for various industrial applications. However, further work is needed to enhance its properties and optimize its use. The production of recombinant enzymes can be carried out in bioreactors following parameter optimization to scale up the yield of MLipA. Protein engineering can further engineer the enzyme to meet the demanding requirements of complex industrial applications. Future research could focus on increasing enzyme thermostability using advanced techniques such as

directed evolution, rational design, and site-directed mutagenesis, as well as molecular dynamics simulations to identify thermolabile areas. Small-scale application studies of organic solvents such as bio-diesel production should also be conducted to assess their utility in real-world industrial applications.

## Supporting information

**S1 Table. List of forward and reverse primers.**
(PDF)

**S2 Table. Components of natural oils used in this study.**
(PDF)

**S1 Fig. Prediction of signal peptide in LipBL by SignalP server.** The signal peptide was predicted to be the first 30 residues LipBL and the cleavage site is between Ala-30 and Ala-31.
(PDF)

**S2 Fig. Displayed all the result of agarose gel electrophoresis using GeneRuler™ 1kb DNA ladder (Fermentas) which denoted as M.** (A) the genomic DNA extracted from *Bacillus licheniformis* IBRL-CHS2. The Lambda DNA/*Hind*III marker (Promega) is denoted as LM. The Genomic DNA is indicated by a white arrow in Lane 1. An agarose gel percentage of 0.7% was utilized for this analysis. (B, C) results of gel electrophoresis showing PCR products amplified with BLF and BLR (B) and the purified LipA gene, identified by the arrow, which is 615 bp in size (C). The samples were run on a 1% agarose gel for electrophoresis. (D) results of colony PCR products from 5 chosen white clones. The LipA gene, measuring 615 bp, is indicated by the arrow. (E) results of agarose gel electrophoresis (1%) following the EcoRI digestion of pGEM-LipA. Lane 1 shows the undigested pGEM- LipA in various conformations, resulting in bands of different sizes. The linear and monomer pGEM- LipA band is highlighted and labeled. In lane 2, digested pGEM- LipA shows the successful cleavage of the LipA gene from the pGEM-T Easy plasmid.
(PDF)

**S3 Fig. Lineweaver plot illustrating the relationship between substrate concentrations and enzyme activity of MLipA.**
(PDF)

**S1 File. The equation for lipase activity (U/mL), relative activity (%), and residual activity (%).**
(PDF)

**S1 Raw images. Original images of gel electrophoresis and SDS-PAGE analysis.**
(PDF)

## Acknowledgments

We extend our sincere gratitude to the 406 Lab, School of Biological Sciences, Universiti Sains Malaysia, and Al-Mustaqbal University for providing the facilities necessary for this study.

## Author Contributions

**Conceptualization:** Ammar Khazaal Kadhim Almansoori, Nidyaletchmy Subba Reddy.

**Data curation:** Ammar Khazaal Kadhim Almansoori, Nidyaletchmy Subba Reddy.

**Formal analysis:** Ammar Khazaal Kadhim Almansoori, Nidyaletchmy Subba Reddy.

**Funding acquisition:** Rashidah Abdul Rahim.

**Investigation:** Ammar Khazaal Kadhim Almansoori.

**Methodology:** Ammar Khazaal Kadhim Almansoori, Nidyaletchmy Subba Reddy.

**Project administration:** Ammar Khazaal Kadhim Almansoori.

**Resources:** Rashidah Abdul Rahim.

**Supervision:** Rashidah Abdul Rahim.

**Validation:** Ammar Khazaal Kadhim Almansoori.

**Visualization:** Ammar Khazaal Kadhim Almansoori.

**Writing – original draft:** Ammar Khazaal Kadhim Almansoori, Nidyaletchmy Subba Reddy, Mustafa Abdulfattah, Sarah Solehah Ismail.

**Writing – review & editing:** Ammar Khazaal Kadhim Almansoori, Nidyaletchmy Subba Reddy, Mustafa Abdulfattah, Sarah Solehah Ismail, Rashidah Abdul Rahim.

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
