## [Decision Letter · Decision Letter 0]

17 Sep 2024

PONE-D-24-37344Characterization of Bacillus licheniformis IBRL-CHS2 Lipase: A novel subfamily 1.4 enzyme.PLOS ONE

Dear Dr. Abdul Rahim,

Thank you for submitting your manuscript to PLOS ONE. After careful consideration, we feel that it has merit but does not fully meet PLOS ONE’s publication criteria as it currently stands. Therefore, we invite you to submit a revised version of the manuscript that addresses the points raised during the review process.

**ACADEMIC EDITOR: **Both reviewers have commented about a major revision are needed before the acceptation of this MS therefore, I invite the authors to revise carefully the MS following closely all recommendations given by the reviewers.

We look forward to receiving your revised manuscript.

Kind regards,

Estibaliz Sansinenea

Academic Editor

PLOS ONE

A clean copy of the edited manuscript (uploaded as the new *manuscript* file).

6. Please include a separate caption for each figure in your manuscript.

Additional Editor Comments:

Both reviewers have commented about a major revision are needed before the acceptation of this MS therefore, I invite the authors to revise carefully the MS following closely all recommendations given by the reviewers.

Reviewers' comments:

Reviewer's Responses to Questions

**Comments to the Author**

1. Is the manuscript technically sound, and do the data support the conclusions?

Reviewer #1: Yes

Reviewer #2: Partly

2. Has the statistical analysis been performed appropriately and rigorously? 

Reviewer #1: No

Reviewer #2: No

3. Have the authors made all data underlying the findings in their manuscript fully available?

Reviewer #1: Yes

Reviewer #2: No

4. Is the manuscript presented in an intelligible fashion and written in standard English?

Reviewer #1: Yes

Reviewer #2: No

5. Review Comments to the Author

Reviewer #1: What is the orginality of the study. Please mention in detailed.

The species name in the enzyme abbreviation should be removed.

Enzyme assay should be detailed.

Statistical analysis should be added.

Reviewer #2: The topic of the manuscript ID: PONE-D-24-37344 is appropriately suited for publication in the PLOS ONE Journal. Some precision, data, and explanation should be also added to improve quality and to facilitate the manuscript’s understanding. The study used classic methods to achieve the results. The latter were discussed well in light of current knowledge and are of relevance to future biotechnological and industrial applications. However, general and specific points need deep attention. Accordingly, in its current form the manuscript needs extensive revisions which I think might be worth considering.

The reviewer has other specific comments/suggestions (authors’ line counts):

- The title should be improved to be more attractive and concise.

- The first sentence in the Abstract should be removed or moved to the introduction section since there are no provided results.

- The acronyms “DMSO”, “SDS”, “PAGE”, “PET”, “OD”, and, “IPTG”, etc, should be spelt out. The abbreviations used in the paper should be firstly spelt out and should be double-checked.

- L25: Please correctly provide the expression unit of the Vmax where “+1” in “min” and “mg” are in superscript.

- Keywords: Please remove “LipAB.licheniformis” and “MLipAB.licheniformis (Mature lipase)”.

- Introduction section should be enhanced and needs revision to properly address the focus of the study. The Authors are invited to provide updated information about the classification lipases from LIPABASE: a database for 'true' lipase family enzymes available at http://www.lipabase-pfba-tun.org

- Introduction section needs enhancement to properly address the focus of the study. Next, since bacterial lipases have been described also in the patent literature it would have been worth to look at such literature as well and not only into the scientific literature.

- The authors are invited to provide the novelty of their work compared to the same one reported elsewhere by Reddy et al., (2016) Trop Life Sci Res. 27(supp1): 145–150. doi: 10.21315/tlsr2016.27.3.20 entitled “Cloning and Expression of a Subfamily 1.4 Lipase from Bacillus licheniformis IBRL-CHS2”!

- In the whole text: Please correct as: “min” instead of “mins” or “minutes”. Next, please correct as: “h” instead of “hours”, “hrs”, or “hour”.

- L145: The word “Stability” is missing in the sub-title. Next, L151: please add “H” after “Tris-”.

- In the whole text, please remove or change the personal words “we” by another expression.

- In Section 2: Please be sure to consistently list manufacturer locations (in this case, by consistently including the country).

- L193-L196: The authors said that “The control reaction, which did not contain any metal ions, was also prepared. The lipase activity in the presence of each metal ion was compared to the control reaction to calculate the relative activity (%).”: To be convinced, the author should mention that the control for the metal cations MUST be measured after dialyzed against used buffer and/or measured in the presence of strong metallic ions chelators (for example 10 mM EDTA or 1 mM EGTA).

- The are some errors in the presentation of citations e.g., “L212: …described in [28].”.

- L232 and elsewhere: The name of restriction enzymes should not be in italic only the restriction enzymes sites (Do not use italics for the first three letters and close up the entire name, e.g., AccI, HaeII. Removal of italics is a change made by IUPAC in 2003.).

- Table 1: How many times the experiments are conducted? Please give the standard error (±SD) values.

- L319 and L582: Please correctly provide the name of “MLipAB.licheniformis's”.

- L501: Please correct as “strain ARM”.

- There are some mistakes in the reference section. Are 123 references necessary? Please remove the oldest references. Please re-check and consult the updated guides to authors, which is available on this journal’s web site, to make them fully comply with the requested style and format of the journal and the name of microorganisms and Latin words should be written in italic some pagination are missing (Ref no.: 76) Please provide correct abbreviation of journal title in accordance with Index Medicus URL: (http://www.bg.ump.edu.pl/czasopisma/medicus.php?lang=eng) and Web of Science: (https://images.webofknowledge.com/images/help/WOS/M_abrvjt.html).

- Finally, some spelling and grammar mistakes should be corrected, and please ask a native speaker to carefully revise the manuscript.

6. PLOS authors have the option to publish the peer review history of their article (what does this mean?). If published, this will include your full peer review and any attached files.

Reviewer #1: No

Reviewer #2: No

---

## [Author Response · Author response to Decision Letter 0]

8 Nov 2024

Responses to reviewers

We sincerely thank the reviewers for their valuable comments and suggestions, which have greatly contributed to the improvement of our manuscript. Below, we have addressed each point in detail.

Table 1: Main Comments and Responses

Main Comment Response Location

What is the originality of the study? Please mention in detail. Thank you for your valuable comment. The originality of this study has been updated in the manuscript. lines 107–133.

The species name in the enzyme abbreviation should be removed. Thank you for your valuable comment. The species name has been appropriately removed from all enzyme abbreviations throughout the manuscript for clarity and consistency. Whole text

Enzyme assay should be detailed. Thank you for your comment. The enzyme assay has been expanded to include detailed procedures, ensuring a comprehensive understanding of the experimental conditions and methodology used. Lines 162-183.

Statistical analysis should be added. Thank you for your comment regarding statistical analysis. I have added a "Statistical Analysis" section to the manuscript. All experiments were performed in triplicate, and the results are presented as mean ± standard deviation (SD). R-values were also calculated to ensure the linearity and reliability of the experimental data. Lines 270-275.

Table 2: Specific Comments and Responses

Specific Comment Response Location

The title should be improved to be more attractive and concise. Thank you for your valuable comment. The title has been updated. Lines 3-4.

The first sentence in the Abstract should be removed or moved to the introduction section since there are no provided results. Thank you for your comment. The first sentence has been removed from the Abstract to improve the flow of information. -

The acronyms “DMSO”, “SDS”, “PAGE”, “PET”, “OD”, “IPTG”, etc. should be spelled out. Thank you for your comment. All abbreviations such as DMSO, SDS, PET, OD, IPTG have been spelled out at their first mention to enhance clarity and ensure consistency throughout the manuscript. Lines 38, 42, 48 and 102.

Please correctly provide the expression unit of the Vmax where “+1” in “min” and “mg” are in superscript. Thank you for your comment. The expression units for Vmax have been corrected, with the appropriate superscripting of "+1" in “min” and “mg,” ensuring that the units are properly formatted. Line 44

Keywords: Please remove “LipAB.licheniformis” and “MLipAB.licheniformis (Mature lipase)”. The keywords “LipAB.licheniformis” and “MLipAB.licheniformis (Mature lipase)” have been removed as per the reviewer’s suggestion to streamline the list of keywords. Lines 50-51

Introduction section should be enhanced and needs revision to properly address the focus of the study. Thank you for your valuable comment. The Introduction has been significantly revised to better align with the focus of the study, providing a clearer and more concise context for the research. Lines 107-133.

The authors are invited to provide updated information about the classification of lipases from LIPABASE. Thank you for your valuable comment. Updated information regarding lipase classification has been added from the LIPABASE database, enriching the background and providing a more comprehensive overview of lipase family enzymes. Lines 72-81.

The authors are invited to provide the novelty of their work compared to the study reported by Reddy et al., (2016). Thank you for your comment. The novelty of our work, in comparison to the study by Reddy et al. (2016), has been addressed and updated in the manuscript. Please refer to Lines (Lines 108-132) where we have detailed how our study overcomes the limitations of the previous work and highlights the unique properties of the enzyme we characterized. Lines 118-132.

Please correct as “min” instead of “mins” or “minutes”, and “h” instead of “hours” or “hrs”. Thank you for your comment. All instances of “mins” and “hours” have been corrected to “min” and “h” to ensure consistency with standard scientific notation. Whole text

The word “Stability” is missing in the sub-title and “H” is missing after “Tris-”. Thank you for your comment. The sub-title has been updated to include the word "Stability," and "H" has been added after “Tris-” to accurately reflect the buffer composition. Lines 192, 193 and 199.

The personal word “we” should be replaced by another expression. Thank you for your comment. All personal expressions using "we" have been replaced with more objective phrasing to adhere to formal scientific writing standards. Whole text

Be sure to consistently list manufacturer locations. Thank you for your feedback. I have ensured that all manufacturer locations are consistently listed throughout the manuscript. Whole text

The authors said that “The control reaction, which did not contain any metal ions, was also prepared. The lipase activity in the presence of each metal ion was compared to the control reaction to calculate the relative activity (%).” To be convinced, the author should mention that the control for the metal cations MUST be measured after dialyzed against used buffer and/or measured in the presence of strong metallic ion chelators (for example 10 mM EDTA or 1 mM EGTA). Thank you for your insightful comment. We have revised the manuscript to clarify the preparation of the control reaction. Specifically, in the Methods section, we now explicitly state that the lipase sample for the control reaction was dialyzed against the buffer used in the experiment to eliminate any trace metal ions that could influence the activity. This ensures that the control is free from any unintended metal ion contamination and allows for accurate comparison with the metal-ion treated samples.

We also acknowledge your suggestion regarding the inclusion of strong metal chelators, such as 10 mM EDTA or 1 mM EGTA, in the control setup. Although chelators were not included in this particular study, we agree that their use in future experiments would further ensure the complete removal of any potential trace metal ions, providing even more robust results. We will incorporate this modification in future studies to validate the observations further.

 Lines 246-250.

Errors in citation presentation (e.g., L212). Thank you for your comment. Citation formatting errors, such as “L212,” have been corrected to ensure proper citation style throughout the manuscript. Line 267

The name of restriction enzymes should not be in italics. Thank you for your comment. The italicization of restriction enzyme names has been removed in accordance with IUPAC guidelines, ensuring correct nomenclature. Whole text

Table 1: How many times were the experiments conducted? Provide standard error (±SD) values. Thank you for your comment regarding Table 1. I have revised the table to include the standard deviation (±SD) values, as requested. The purification experiments were conducted in triplicate, and the updated table now reflects the appropriate means and SD values for each parameter. Lines 350-352

L319 and L582: Please correctly provide the name of “MLipAB.licheniformis's”. Thank you for your feedback. The name “MLipAB.licheniformis's” has been corrected to align with proper scientific terminology. Whole text

L501: Please correct as “Strain ARM”. Thank you for your comment. The term “strain ARM” has been corrected for clarity and accuracy. Line 551

Please remove the oldest references if not necessary. Thank you for your feedback. The older references were selected because they focus specifically on Bacillus lipases, which are directly relevant to my study. There are limited new references specifically on Bacillus lipases, and the newer studies often involve lipases from different sources. To maintain a logical and meaningful comparison, I prioritized older studies on Bacillus lipases, which align more closely with my research focus. -

Please re-check and consult the updated guides to ensure compliance with journal style. Thank you for your feedback. I have reviewed the manuscript in accordance with the updated guidelines to ensure compliance with the journal’s style. Whole text

Ref no.: 76: Provide the correct journal abbreviation in accordance with Index Medicus and Web of Science. Thank you for your feedback. The corrected journal abbreviation is verified per Index Medicus and Web of Science. Line 1134

Some spelling and grammar mistakes should be corrected. Thank you for your feedback. I have carefully reviewed the manuscript and corrected the spelling and grammar mistakes as suggested. Whole text

Data availability response

Thank you for your comments regarding the Data Availability Statement. The manuscript contains all the essential data needed to replicate the findings of this study. However, due to ongoing enhancements, including the application of peptides to improve lipase stability, some original data is not yet fully formatted for public sharing.

We plan to share all relevant data in future publications once the enhancements are complete. In the meantime, we kindly request that the data provided within the manuscript be considered sufficient to replicate the current study’s results. Should the editor or any reviewers have specific questions or require access to the data during the review process, we are happy to provide these privately upon request.

---

## [Editor Report · Decision Letter 1]

13 Nov 2024

Characterization of a novel subfamily 1.4 Lipase from Bacillus licheniformis IBRL-CHS2: Cloning and expression optimization

PONE-D-24-37344R1

Dear Dr. Abdul Rahim,

We’re pleased to inform you that your manuscript has been judged scientifically suitable for publication and will be formally accepted for publication once it meets all outstanding technical requirements.

Kind regards,

Estibaliz Sansinenea

Academic Editor

PLOS ONE

Additional Editor Comments (optional):

The authors have done all changes suggested by both reviewers according to their comments improving the Ms, therefore, the MS can be accepted in the current form.
---

## [Editor Report · Acceptance letter]

6 Dec 2024

PONE-D-24-37344R1 

PLOS ONE

Dear Dr. Abdul Rahim, 

I'm pleased to inform you that your manuscript has been deemed suitable for publication in PLOS ONE. Congratulations! Your manuscript is now being handed over to our production team.

Kind regards, 

on behalf of

Dr. Estibaliz Sansinenea 

Academic Editor

PLOS ONE